# Synthesis and Characterization of Magnetic Molecularly Imprinted Polymer Sorbents (Fe_3_O_4_@MIPs) for Removal of Tetrabromobisphenol A

**DOI:** 10.3390/ijms26167686

**Published:** 2025-08-08

**Authors:** Clarissa Ciarlantini, Susanna Romano, Gian Marco Amici, Elisabetta Lacolla, Iolanda Francolini, Anna Maria Girelli, Andrea Martinelli, Antonella Piozzi

**Affiliations:** 1Department of Chemistry, Sapienza University of Rome, P. le Aldo Moro 5, 00185 Rome, Italy; clarissa.ciarlantini@uniroma1.it (C.C.); marcogiam123@gmail.com (G.M.A.); elisabetta.lacolla@uniroma1.it (E.L.); iolanda.francolini@uniroma1.it (I.F.); annamaria.girelli@uniroma1.it (A.M.G.); andrea.martinelli@uniroma1.it (A.M.); 2Department of Industrial, Electronic and Mechanical Engineering Roma Tre University, Via Vito Volterra 62, 00146 Rome, Italy; susanna.romano@uniroma3.it

**Keywords:** magnetic nanostructured sorbents, magnetic molecularly imprinted polymer systems, flame retardants, tetrabromobisphenol A, pollutants removal

## Abstract

Tetrabromobisphenol A (TBBPA) is a flame retardant widely added to polymer products. Successful isolation of target analytes from complex natural matrices relies on extraction materials that can selectively interact with the analytes. In this context, the use of magnetic nanostructured adsorbents, such as magnetic molecularly imprinted polymer systems (MMIPs), can play a key role in both selective matrix–analyte interactions and separation processes. Here, to achieve different TBBPA loadings, Fe_3_O_4_ nanoparticles (NPs) were coated with chitosan (CS) or (3-aminopropyl) triethoxysilane (APTES). Moreover, to further promote template–NP interactions and modulate the polymeric shell thickness of MMIPs, 3,4-dihydroxyhydrocinnamic acid (HC) was covalently bonded in different amounts to APTES-functionalized MNPs. Thermal, SEM, and elemental analyses showed a different coating degree of the nanocomposites (Fe_3_O_4_@CS-MIP size d = 77 nm and Fe_3_O_4_@APTES-MIP d = 20 nm). In addition, it was confirmed that the adsorption mechanism of TBBPA on Fe_3_O_4_@APTES-HCX-MIPs was due to specific interactions between the systems and the analyte, unlike non-imprinted analogs (MNIPs). Among the developed systems, the Fe3O4@APTES-HC0.7-MIP sample showed the best extraction efficiency (85%) associated with good discharge efficiency (70%). Furthermore, this nanocomposite displayed high selectivity towards TBBPA (ε > 1) and good extraction efficiency in three consecutive cycles (67%), demonstrating great potential in the environmental field.

## 1. Introduction

Halogenated flame retardants (HFRs) are substances used in common flammable objects or materials to reduce smoke development and contain flame propagation in the event of a fire. Many of these compounds are associated with adverse health effects, including cancer, reproductive toxicity, immunotoxicity, endocrine disruption, and adverse effects on fetal and child neurodevelopment [1]. Among the most widely used flame retardants is tetrabromobisphenol A (TBBPA), which is widely employed in the production of polycarbonate, phenolic, and epoxy resins, as well as in the engineering plastics industry [2]. Environmental studies have revealed that TBBPA can enter the environment during production, handling, and end-product use, as well as disposal in the form of micro- or nano-plastics [3]. TBBPA is currently present in air, water, soil, indoor dust, sediment, and sewage sludge from various locations. The main exposure routes for the population include diet, ingestion, and skin contact with dust. As a result, TBBPA can be detected in human body fluids, such as in breast milk and maternal blood serum/cord blood. Cariou et al. and Kim and Oh demonstrated that TBBPA levels in blood serum can reach 93.22 ng/g lipid weight (lw) in adults and 457.4 ng/g lw in infants [4,5]. To date, TBBPA has not been linked to the onset of diseases or medical conditions. However, its toxicity has been extensively studied and reported by Zhou et al., who collected most toxicological studies and evaluated the doses of TBBPA exposure that may affect human health [6]. For these reasons, TBBPA is a persistent pollutant that requires its determination in complex matrices. The correct isolation of target analytes from complex natural matrices relies on innovative analytical techniques and extraction materials that interact with the analytes selectively.

As regards extraction techniques, in recent years, the application of solid phase micro-extraction (SPME) is increasingly replacing the use of liquid–liquid extraction (LLE) [7,8]. Its wide use in the analytical field derives from the reduced amount of solid used for the extraction (minimal consumption of adsorbent) and from the ease of implementation of the technique, combined with the economy in terms of time and volume of solvent used. Coupling high-selectivity extractive materials in analyte capture with SPE has resulted in an ideal setup for processing complex matrices of real samples [9]. Nowadays, the most widely used materials are polymeric or silica-based sorbents and, more recently, nanostructured sorbents [10,11]. The use of nanostructured sorbent materials has as its main advantage the high surface/volume ratio, which can promote interactions between the system and the pollutant by facilitating its capture. For example, Alsalem et al. developed functionalized mesoporous silica nanotubes (FMSNTs) for adsorption of TBBPA from contaminated water, using (3-aminopropyl) triethoxysilane (APTES) to achieve a response surface technology using SPE methodology [12]. In addition, Chen et al. synthesized melanin nanoparticles with various functional groups introduced through a self-assembly strategy that allowed them to quantify TBBPA and TBBPS within food samples [13].

However, some of them lacked selectivity, especially when complex matrices were processed, leading to the co-extraction of interfering substances. In recent years, molecular imprinting technology (MIT) has been considered an interesting method to produce artificial receptors obtained through memory of the size, shape, and functional groups of template molecules [14,15]. In addition, these matrices have proven ideal for processing real samples through DμSPE (dispersive micro-solid phase extraction) due to their specific recognition properties based on the formation of covalent or ionic complexes between functional monomers and model molecules [16]. These complexes are then immobilized by copolymerization, taking advantage of the presence of cross-linking agents. Once polymerization is complete, the template molecules are removed, providing binding sites with complementary shapes, sizes, and functionalities to the target template.

Compared with traditional media used for separation and analysis, MIPs are characterized by good mechanical/chemical stability, high specificity, low cost, ease of preparation, and reversible adsorption/release of the target molecule, making them ideal materials for sample separation and analysis [17,18]. Conversely, some drawbacks, such as slow mass transfer, irregular shape, incomplete template removal, poor site accessibility, or heterogeneous distribution of MIP binding sites, have limited their application. For these reasons, surface molecularly imprinted polymers (SMIPs) have been developed in recent years to overcome the drawbacks of bulk polymerization applied to MIP production [19].

This strategy allows the immobilization of recognition sites on the surface of a solid matrix, which has characteristic properties able to improve the extraction and determination step of the pollutant [20,21]. In this field, the use of magnetic MIPs (MMIPs) based on iron oxides (Fe_3_O_4_) is attracting research interest [22]. These systems are easy to produce, inexpensive, and allow the recovery of the system by means of a magnetic field, which is a key property in the production steps and subsequent sample processing [23]. In their synthesis, it is imperative to obtain magnetic nanoparticles (MNPs) that are stable under different conditions and especially capable of allowing deposition of the polymer layer required to produce specific cavities in the in situ polymerization step.

The choice of nanoparticle production technique plays a key role in determining the size of NPs [24]. The co-precipitation method has proven to be one of the most widely applied methods due to the high magnetite yield achieved by a simple procedure [25,26]. However, attention should also be paid to exploring different surface modification strategies required to enable the formation of MMIPs [27]. The need to modify the surface of MNPs is due to the low number of functional groups present on their surface, which limits the success of polymer layer deposition during the in situ polymerization step for MMIP production. In addition, the introduction of surface coatings with small molecules or polymeric molecules may limit the aggregation phenomenon typical of such systems [28].

In the literature, different Fe_3_O_4_-NP coatings have been studied to reduce this phenomenon due to the short-range van der Waals attractive forces and residual magnetization forces between the particles [29]. For example, Apriceno et al. pointed out the possibility of making chitosan (CS)-based coatings of magnetite NPs, limiting the aggregation phenomenon and increasing the catalytic activity of the realized system [30]. Since magnetic NPs are versatile systems, their applications in the production of MIPs for the sensor field [31] and in the environmental field have been highlighted in the literature. For example, Shao et al. made MMIPs from Fe_3_O_4_ whose molecularly imprinted cover shell was prepared by the combination of click chemistry and RAFT polymerization [32]. These systems showed high selectivity in capturing TBBPA and an extraction capacity of the pollutant around 65%.

However, the production of a selective surface coating capable of capturing specific molecules within complex matrices and with high performance still represents a major environmental challenge. To the best of our knowledge, there are few studies in the literature on the fabrication of engineered coatings able to specifically interact with the analyte to be captured [33,34]. In the preparation of MIPs, attention should be paid to the realization of the pre-polymerization complex between the template and the functionalized NPs [35,36]. The use of NPs modified with groups capable of interacting with the template allows for more cavities and a better distribution of those cavities within the polymer surface resulting from the in situ polymerization phase.

For these reasons, in this work, MMIP systems based on differently functionalized iron oxides were prepared for selective capture of TBBPA in real and complex matrices. In particular, the effect of two surface coatings of magnetite NPs, one polymeric and other non-polymeric, on the extraction capacity of developed MMIP systems was studied. In the case of the polymer coating, a polysaccharide such as CS was chosen, which contain many amino and hydroxyl groups that could be exploited both for its physical immobilization on the surface of magnetite NPs and to promote interaction with the target analyte during the formation of MIPs. In addition, CS is known for its high adsorptive properties, which are widely exploited in the analytical field [37,38,39].

Since the size of the nanostructured system can also affect the performance of the adsorbent, a small molecule containing only amino groups, such as (3-aminopropyl) triethoxysilane (APTES), covalently bonded to MNPs, was also used to make functionalized magnetic systems. In addition, to further favor the interaction of the matrix with the template, thus allowing the formation of a higher number of homogeneously dispersed cavities and at the same time modulating the thickness of the polymer shell during MIP synthesis, an antioxidant molecule such as 3,4-dihydroxyhydrocinnamic acid (HC) was covalently bonded to the functionalized NPs using three different molar ratios between the amino group of APTES and the carboxylic group of HC (1:0.3, 1:0.7, and 1:1.0). MMIPs, in fact, were obtained by radical polymerization of methacrylic acid (MAA) and ethylene glycol dimethacrylate (EGDMA) adsorbed onto the functionalized nanosystems in the presence of the template. To the best of our knowledge, there is no study in the literature regarding the introduction of antioxidant molecules covalently bonded to inorganic matrices aimed at improving the interaction of NPs with the template and at the same time controlling the size of nanoparticles.

All systems were characterized by infrared spectroscopy (FTIR), thermogravimetric analysis (TGA), elemental analysis, and electron microscopy (FESEM) coupled to EDX analysis. The effective functionalization of NPs with CS and APTES was confirmed by the ninhydrin (NHN) test, which allowed us to determine the surface concentration of the amino groups, while the introduction of the antioxidant molecule was verified by the DPPH test, also highlighting the antioxidant activity of these matrices. The extraction capacity of the systems towards TBBPA was studied by HPLC-UV. The adsorption isotherms and the adsorption kinetics were also studied. Finally, the most promising systems were subjected to reuse cycles and their selectivity in capturing TBBPA was evaluated with respect to two similar pollutants such as 2,4-dibromophenol (2,4DBP) and 4-bromophenol (4BP).

## 2. Results and Discussion

The development of matrices able to interact selectively and with high affinity with pollutants within real matrices is of fundamental importance for the remediation of polluted habitats. Molecularly imprinted polymer technology coupled with the DμSPE technique can overcome the limitations imposed by commonly used adsorbents [16]. Furthermore, the production of molecularly imprinted polymers (MIPs) on the surface of magnetic particles has enabled the development of many innovative systems [40,41]. Magnetic nanoparticles (Fe_3_O_4_) in core–shell-based structures provide a high surface area for anchoring MIPs, and the surface can also be activated/functionalized with different molecules that can promote interactions with the template, favoring the formation of specific cavities.

Several MMIP systems have been developed over the years that have evaluated different modifications of the surface of magnetite nanoparticles. For example, Wu et al. made water-compatible temperature- and magnetic-field-responsive MIPs for the selective recognition and extraction of bisphenol A, using Fe_3_O_4_ nanoparticles as the magnetic core, SiO_2_ nanoparticles as the coating materials, methacrylic acid (MAA) as the functional monomer, and NIPAM as the temperature-sensitive functional co-monomer [42]. Instead, Pan et al. synthesized MMIP systems based on attapulgite/Fe_3_O_4_ particles to selectively recognize 2,4-dichlorophenol (2,4DCP) [43]. The polymerization step, carried out using MAA and EGDMA as the monomer and cross-linking agent, respectively, resulted in an innovative adsorption matrix with high selectivity in capturing 2,4DCP compared to similar contaminants.

In this work, two different types of coatings were used: the first was polymeric (based on CS), and the second was based on small molecules such as APTES. Applications of CS in the production of magnetic MIPs are reported in the literature. For example, Huang et al. reported the production of MIP systems based on CS modified with beta-cyclodextrins (β-CDs) anchored on the surface of magnetite for the capture of bisphenol A (BPA) [44]. This study demonstrated how the presence of the -OH and -NH_2_ groups of CS combined with the presence of the hydrophobic cavity typical of β-CDs effectively improved the selectivity of MMIP systems.

APTES has also been widely used in the production of MIPs, as reported by Mehdipour et al., who developed magnetic MIP systems functionalized with APTES for the capture of organophosphorus pesticides [45], or by Liu et al., who obtained MMIPs for the chiral separation of the tryptophane enantiomers [46].

However, no studies comparing these two different magnetite surface coatings for the production of MIP systems for environmental applications have been reported in the literature. In addition, since the number of functional groups introduced by APTES is low and the interaction of the nanosystems with the template could be affected, in this study, APTES-functionalized MNPs were further modified by introducing 3,4-dihydroxyhydrocinnamic acid (HC), which can interact with the template through aromatic stacking interactions. The choice to bind an antioxidant to the MNPs was also dictated by the possibility of limiting the thickness of the polymeric shell during MIP formation by controlling the radical polymerization of MMA and EGDMA monomers. To this end, three different molar ratios between the amino groups on the MNP surfaces and the carboxylic groups of HC (1:0.3, 1:0.7, and 1:1.0) were used. To confirm the effectiveness of HC introduction and compare CS- and APTES-containing systems with each other, MNPs were subjected to preliminary TBBPA extraction measurements before MIP formation. After 24 h of contact with the flame retardant, the HC-containing systems showed higher binding capacity (Q = 0.08 mg/g) than the system functionalized with APTES (Q = 0.04 mg/g) and the CS-coated system (Q = 0.05 mg/g), confirming the efficacy of this strategy. Next, the MMIP systems were implemented by radical polymerization in the presence of the functionalized MNPs, using MAA and EGDMA as the functional monomer and cross-linking agent, respectively, and BPO as the initiator. Figure 1 shows the synthesis scheme of the developed systems.

### 2.1. Infrared Spectroscopy

FT-IR analysis was used to qualitatively confirm the coating of MNPs with CS, their functionalization with APTES and HC, and finally the synthesis of the MMIP systems. Figure 2A shows the spectra of CS, magnetite nanoparticles, and MNPs after coating with the polysaccharide. The CS spectrum, as reported by Brugnerotto et al., showed the following characteristic absorptions: broad absorption between 3500–3000 cm^−1^, corresponding to -OH and -NH stretching frequencies; C-H stretching in the range 2920–2875 cm^−1^; the C=O stretching of acetylated groups (amide I) at 1650 cm^−1^; N-H bending of primary amine together with N-H in the plane deformation (amide II) and C-N stretching at 1560 cm^−1^; finally, absorptions due to the pyranose ring in the range 1150–1000 cm^−1^, particularly C-O-C and C-O-H stretching at 895 cm^−1^ [47].

The spectrum of magnetite (Fe_3_O_4_) showed the characteristic bands related to the hydroxyls on the surface. In particular, the absorption in the range 1050–950 cm^−1^, due to the vibration of the Fe-OH bond; H-O-H bending, which results in absorption between 1650–1340 cm^−1^; and, finally, a very broad band between 3650–2500 cm^−1^ characteristic of stretching of the O-H bond [48]. The polymer coating was confirmed by the increase in bands in the region between 1700–1200 cm^−1^, related to amide I and amide II vibrations of CS, as well as the presence of a new band at 1150–1000 cm^−1^ due to the pyranose ring. As observed with CS, functionalization with APTES was also qualitatively confirmed by FT-IR.

Figure 2B shows the spectra of APTES, magnetite nanoparticles, and MNPs after functionalization with APTES and HC. As an example, only the spectrum of MNPs functionalized with the lowest APTES:HC ratio (Fe_3_O_4_@APTES-HC0.3) is shown in Figure 3B. In the spectrum of the pristine APTES, the C-H stretching of propyl groups could be detected around 2950–2800 cm^−1^, as well as the stretching bands of the Si-O-C and C-N bonds and bending deformations of -NH_2_ at 1070, 940, and 750 cm^−1^, respectively. The functionalization of magnetic nanoparticles was confirmed by the appearance of C-H bond stretching absorptions (2950–2800 cm^−1^) and the characteristic peak of symmetric and asymmetric Si-O-C bond stretching, centered at about 1000 cm^−1^, as reported by Maity et al. [49]. Regarding the introduction of HC, it could be seen that, compared with the spectrum of MNPs coated with silane alone, the spectrum of magnetite functionalized with the antioxidant exhibited the characteristic absorptions of the hydroxyl groups and, most importantly, the appearance of absorption related to the stretching of the C=C bond of the aromatic ring and the amide C=O bond at about 1600 cm^−1^.

Finally, confirmation of in situ polymerization on the surface of magnetite functionalized with CS, APTES, and APTES-HC was obtained by analyzing the IR spectra of the MMIP and MNIP samples. As an example, Figure 2C shows the spectra of the MNIP and MMIP obtained by employing the Fe_3_O_4_@APTES-HC0.3 system compared with those of the samples before polymerization. The magnetite surface coating formed by the MAA-EGDMA polymer was characterized by the following absorptions: band between 3600–2500 cm^−1^, characteristic of O-H bond stretching; band at 1729 cm^−1^ due to C=O bond stretching of the ester; bands in the 1500–1300 cm^−1^ region, related to the CH_3_ and CH_2_ bending vibrations; band at 1200–1180 cm^−1^, due to the C-O-C stretching of the ester bond. No differences were observed in the absorptions between the MMIP and MNIP systems, indicating the effectiveness of the washing step in removing the template used to produce the specific cavities. These absorption bands agree with those highlighted in the literature [43,44,50].

### 2.2. Field Emission Scanning Electron Microscopy (FE-SEM)

The surface morphology of the prepared systems was investigated by FE-SEM analysis. As shown in Figure 3i, it was possible to confirm the spherical shape of the magnetite nanoparticles after the synthesis process, with a diameter of about 15 nm (Figure 3ii), as reported by Zhang et al. [51]. After coating with CS, an increase in the size of nanoparticles was observed (d = 40 nm), with particles partly embedded in the polymer layer. This result agrees with that found by Apriceno et al., who used MnFe_2_O_4_ magnetic nanoparticles coated with CS to immobilize laccase for the removal of contaminants [30]. APTES-coated MNPs instead showed a morphology almost unchanged compared to that of the naked magnetite (Figure 3(iC)) and size of about 20 nm even after the introduction of the antioxidant molecule at different concentrations (see Figure 3(iD),ii).

To evaluate the effect of the in situ polymerization of the acrylic monomer on the morphology of the nanoparticles, as an example, the micrographs of the MMIPs obtained from the Fe_3_O_4_@CS, Fe_3_O_4_@APTES, and Fe_3_O_4_@APTES-HCAF0.3 systems are reported in Figure 4i, while the average diameters of all the developed systems are reported in Figure 4ii.

From the micrographs, it was possible to notice that, in general, the in situ polymerization phase did not cause aggregation of the nanoparticles, which maintained a spherical morphology. However, in the Fe_3_O_4_@CS-MIP system, an increase in the size of MNPs was noted, reaching a diameter of approximately 80 nm. The larger size of the nanocomposites containing CS, compared to those with APTES (diameter around 35 nm), suggests a greater coating thickness. This is likely due to the functional groups in the polysaccharide structure, which interact more effectively with the acrylic matrix and promote the formation of the polymer shell.

This result was confirmed by Hashemi et al., who reported a diameter of approximately 95 nm for CS-coated magnetite following the MIP preparation process [52].

The presence of antioxidant molecules on the APTES-containing nanoparticles, particularly at higher concentrations, influenced the thickness of the acrylic polymer shell, decreasing it, even in the absence of a template. This allowed the formation of more homogeneous systems with a high surface/volume ratio. This result confirmed the ability of the antioxidant to capture radical species during the polymerization phase [53].

### 2.3. Determination of the Concentration of Amino Groups

The ninhydrin assay was used to obtain a quantitative assessment of the degree of magnetic nanoparticle coating. In the literature, several investigations have reported the appearance of the band at about 570 nm related to the production of a colored complex resulting from the reaction between ninhydrin and primary amino groups. This absorption can be used for quantitative analysis of the surface functionalization of systems with molecules containing amino groups [54,55]. In this study, this analysis made it possible to detect the concentration of surface amino groups (mM) per g of the sample tested, which is directly related to the amount of APTES and CS present on the surface of the nanocomposites. The Fe_3_O_4_@CS system showed a surface concentration of amino groups equal to 9.1 ± 0.5 mM/g, in agreement with the result found by Ziegler-Borowska et al. [56]. As for the Fe_3_O_4_@APTES sample, a significantly lower surface concentration of amino groups was found than that of the system containing CS (0.13 ± 0.02 mM/g). This difference was attributable to the polymeric nature of CS, given its high molecular weight.

### 2.4. Determination of HC Content and Antiradical Capacity of MNPs by DPPH Method

The quantification of the Fe_3_O_4_@APTES nanocomposite functionalization with HC, in the three prepared molar ratios (APTES:HC 1:0.3, 1:0.7, and 1:1.0), was performed using the DPPH assay. The introduction of this molecule was necessary to favor matrix–template interaction and modulate the formation of the pre-polymerization complex, limiting the thickness of the shell.

The amount of HC bound to MNPs was expressed as the antioxidant concentration (μM) per g of the sample tested. As the APTES:HC ratio increased, an increase in surface concentration was observed. Specifically, samples Fe_3_O_4_@APTES-HC0.3, Fe_3_O_4_@APTES-HC0.7, and Fe_3_O_4_@APTES-HC1.0 had surface concentrations of HC equal to 7.5, 10, and 12.5 μM/g, respectively. Furthermore, for these systems, it was also possible to determine the antiradical capacity (*AC*%), which increased with increasing coating. In particular, the samples Fe_3_O_4_@APTES-HC0.3, Fe_3_O_4_@APTES-HC0.7, and Fe_3_O_4_@APTES-HC1.0 showed an *AC* (%) of 34 ± 2, 52 ± 3, and 66 ± 5%, respectively. This parameter represented the ability of these systems to limit the formation of the imprinted shell, given their ability to capture radicals during the production phase of the MIP and NIP systems. These results confirmed the data obtained from FE-SEM analysis, which showed a decrease in the diameter of the imprinted nanoparticle systems as the molar ratio between APTES:HC increased.

### 2.5. Elemental Analysis

By determining the elemental compositions of the nanocomposite masses, expressed as a percentage of carbon and nitrogen, it is possible to confirm the surface functionalization of the coated magnetic systems [30,54,57]. In particular, for MNPs functionalized with CS and APTES-HC, the ratio between experimental nitrogen and carbon was taken into consideration and compared with the theoretical ratio (Table 1).

The results confirmed the functionalization of the samples, since the experimental N/C ratio was very similar to the theoretical one. Observing the N/C ratio for the samples functionalized with the antioxidant molecule, it was possible to notice that as the content of bound antioxidant increased, the experimental N/C ratio decreased, in agreement with the theoretical data.

For the imprinted systems, it was decided to evaluate the C content per mg of tested matrix (Table 2). In this way, it was possible to quantify the coating due to the synthetic polymer.

The elemental analysis showed no difference in the degree of coverage between the MMIP and MNIP systems, demonstrating the effectiveness of the TBBPA washing procedure used for MMIP production. Analysis of the MMIP and MNIP systems obtained from CS and APTES confirmed a higher degree of coating for CS-functionalized matrices. This result agrees with FE-SEM measurements showing a larger diameter for this nanosystem and with data from the ninhydrin assay indicating a higher concentration of amine groups for the Fe_3_O_4_@CS system. The decrease in the carbon content (%), and thus in the degree of coating, of the Fe_3_O_4_@APTES systems containing the antioxidant as its concentration increased confirmed the ability of HC to modulate the thickness of the coating through the capture of free radicals produced during the in situ polymerization phase, as also demonstrated by measurements of the antiradical capacity of these systems.

### 2.6. Thermogravimetric Analysis

In addition to elemental analysis, the samples were subjected to thermogravimetric analysis, through which the presence of the coatings on the prepared samples could be further confirmed. The thermal stability of the nanosystems depends on the interactions that occur between the inorganic core and the surface organic coating layer. Therefore, the degradation of the organic part appears to be directly related to the functionalization of the surface [58]. The percent weight loss of the nanosystems and their degradation temperature (T_d_) are shown in Table 3, while thermograms of naked magnetite and after coating with CS, APTES, and APTES-HC0.3, the latter reported as an example, are shown in Figure 5A.

The naked magnetite showed a slight degradation up to 130 °C of about 4% wt, attributable to the presence of strongly bound water, as also reported by Zhang et al. [59]. The higher weight loss of the Fe_3_O_4_@CS sample (13%) compared with the Fe_3_O_4_@APTES sample (9%) confirmed the higher coating of MNPs with the polysaccharide. In addition, while the introduction of the antioxidant molecule reduced the shell thickness of the Fe_3_O_4_@APTES-HC samples by about 3%, no clear trend related to the change in HC concentration was observed.

The thermal stability of developed systems, on the other hand, increased with the coating/functionalization of MNPs. In particular, the T_d_ of the Fe_3_O_4_@CS system was higher (300 °C) than that of CS alone (290 °C), a value determined in our previous work [60]. This increase was attributable to the interactions created between the inorganic core and the CS. The formation of covalent bonds between magnetite and APTES justified the further increase in the T_d_ of the Fe_3_O_4_@APTES nanosystem (321 °C). Also, Hozhabr Araghi et al. observed that the temperature for breakdown of APTES grafted on mesoporous-silica magnetic nanoparticles (T_d_ = 342 °C) was much higher than the boiling point of pure APTES (T_eb_ = 217 °C) [57].

For the HC-containing samples, the significant increase in T_d_ values with an increasing amount of introduced antioxidant was due to the formation of aromatic stacking interactions between the HC rings, which further improved the thermal stability of the systems.

By analyzing the produced MIP and NIP systems (Table 3), it was possible to observe a significant increase in the % weight loss compared to the unpolymerized materials, demonstrating the formation of a polymer shell of considerable thickness. As an example, the thermogravimetric curves of the samples obtained from the Fe_4_@APTES-HC0.3 system are shown in Figure 5B. Obviously, more significant degradation was found for the MIP and NIP systems obtained starting from the Fe_3_O_4_@CS system than for those obtained starting from APTES, given the polymeric nature of CS. This result was in agreement with SEM data, which highlighted a larger diameter of this nanosystem, and also with the elemental analysis measurements. The same results were also obtained by Mehdipour et al. and Zhang et al., who observed greater weight loss for MIP systems made with CS-coated magnetite than those made with APTES [45,59].

Finally, for the systems containing HC, there was not significant different weight loss than for systems containing only APTES. Instead, an increase in T_d_ was observed with increasing antioxidant concentration, which confirmed the stabilizing effect of HC even after the in situ polymerization step.

### 2.7. Study of TBBPA Adsorption

Adsorption kinetics of TBBPA on the developed matrices were performed using a specific concentration of flame retardant (4 ppm) and pH = 5. The choice of pH was mainly due to the stability of TBBPA. As reported by Yu et al., TBBPA is degraded to mono- or divalent anions when pH > 8.0, resulting in weak binding to the recognition sites [61]. Furthermore, at a pH below the pKa of CS (equal to 6.5 [62]), more interactions between the matrix and the pesticide could be established [17]. Finally, for APTES-containing samples, use of an acidic pH could allow good capture of TBBPA. As confirmed by Otalvaro et al., a higher adsorption capacity was found for the systems containing APTES towards a pesticide such as 2,4-dichlorophenoxyacetic acid (2,4-D), for a pH around 4.5 [63].

Figure 6A shows the adsorption kinetics of the MIP and NIP samples made with Fe_3_O_4_@CS and Fe_3_O_4_@APTES. As can be observed, with increasing contact time, the rate of the pesticide adsorption increased rapidly in the first 30 min and then decreased, reaching equilibrium at different times. This behavior was directly related to the presence of a diverse number and typology of binding sites for TBBPA available on the surface of the developed MIP and NIP matrices. In fact, in the initial phase of the adsorption process, many active binding sites were available on the surface of the systems, particularly on the MIP systems, but as the adsorption progressed, these sites were gradually occupied by TBBPA.

Looking at the curves specifically, it can be seen that in the case of the CS-coated nanocomposite, no clear differences were visible between the MMIP and MNIP systems. This behavior could be attributed to the fact that TBBPA, being able to penetrate the polymer layers of the polysaccharide, was then partially blocked by the next polymer layer following the formation of MIPs. This could have led to the reduced formation of specific cavities determined by the lack of diffusion of the flame retardant towards the surface of the system. This effect was less present in MMIPs and MNIPs obtained from Fe_3_O_4_@APTES, in which a covalent bond was formed between magnetite and a low-molecular-weight molecule such as silane. In these samples, a higher Q value was observed for the MMIP system than for the MNIP one. This result confirmed the formation of specific cavities in the MMIP capable of capturing the flame retardant [64]. Moreover, further confirmation of the presence of specific cavities in the Fe_3_O_4_@APTES system was given by reaching the plateau in a shorter time than the non-imprinted system, as also found by Hashemi et al. [52].

As previously mentioned, to promote the formation of the pre-polymerization complex, improve the extraction efficiency of the APTES-containing nanocomposites, and control the polymer shell thickness during radical polymerization, an antioxidant molecule (HC) was introduced. Figure 6B shows the adsorption kinetics, up to 24 h, of MMIP and MNIP obtained using different molar ratios of APTES:HC (1:0.3, 1:0.7, and 1:1.0). It was verified that only in the case of the Fe_3_O_4_@APTES-HC0.7-MIP sample was a higher Q value obtained compared to the system containing only APTES. Probably, in the case of the lowest HC concentration, the number of specific cavities formed, able to selectively capture TBBPA, was too low, while, in the case of the highest antioxidant concentration, the capture of radicals formed during the MMA polymerization step limited the formation of the imprinted shell, which was similar in thickness to that of the Fe_3_O_4_@APTES-HC0.7-MIP sample, as demonstrated by FE-SEM measurements. The best performance of this system was also confirmed by the absorption efficiency measurements (Appendix A). In fact, the Fe_3_O_4_@APTES-HC0.7-MIP sample showed the highest AE (%) value, equal to 85% while in the other samples containing the antioxidant this value remained around 50%.

These results could also be attributed to the better dispersibility of the nanoparticles in the Fe_3_O_4_@APTES-HC0.7-MIP sample compared to that of the other two systems. Indeed, this sample remained stable for at least 24 h, showing only a slight aggregation of the nanoparticles after 48 h (see Appendix A). With increasing HC concentration, aggregation phenomena were already visible after 24 h (Appendix A). Although the adsorption kinetics were performed while keeping the samples under stirring, the reduced surface/volume ratio due to MNP aggregation could affect the availability of the specific cavities for TBBPA. In fact, although a small variation in adsorption efficiency was observed for the Fe_3_O_4_@APTES-HC0.7 systems at 48 h (AE = 80%), a greater reduction was observed for the Fe_3_O_4_@APTES-HC1.0 system (AE = 38%).

The Fe_3_O_4_@APTES-HC0.7-MIP system was also characterized by a good desorption capacity DE (%) in the first cycle, with a value around 70%. Finally, considering the affinity (α), i.e., the measure of the recognition properties towards the analyte of the imprinted and non-imprinted polymers, an α value higher than 1 was observed for all the developed matrices. This indicated a higher recognition capacity of the MIP systems for the analyte compared to the non-imprinted analogs. However, only in the Fe_3_O_4_@APTES-HC0.7-MIP sample, this parameter assumed a value around 2.0, indicating the high affinity of this system for TBBPA and therefore its high potential for use in the environmental field (Appendix A).

The adsorption process of an adsorbate on a porous material occurs through a series of steps: mass transfer from the fluid phase to the surface through the boundary layer, diffusion inside the pores of the material, and adsorption on the internal surface of the adsorbent [65]. Several mathematical models are used to describe the adsorption process of an adsorbent on a surface in terms of equilibrium and kinetic behavior. While to investigate the interaction between adsorbate and sorbent, Langmuir or Freundlich type adsorption isotherm models are considered, to examine the adsorption mechanism, the kinetic data are analyzed using pseudo-first-order and pseudo-second-order rate equations. The mathematical model that seems to best describe the adsorption mechanism of adsorbates on MIP systems is the pseudo-second-order model, as evidenced in several works in the literature [66,67]. As reported by Shao et al., for MMIP systems developed for TBBPA capture, this model confirmed that chemical and surface adsorption reactions probably drive the rate-limiting step [32].

The adsorption kinetic data of all the systems developed in this work were analyzed using both pseudo-first-order and pseudo-second-order rate equations. However, only the data for the Fe_3_O_4_@APTES-HC0.7-NIP/MIP sample were reported, as this system proved to be the most promising in terms of TBBPA capture (Figure 6C,D). The data relating to the kinetic parameters, reported in Table 4, showed that the NIP sample followed pseudo-first-order kinetics, while the MIP sample, as previously mentioned, followed pseudo-second-order kinetics. The adsorption mechanism found for the NIP sample was due to the absence of surface cavities that limited the TBBPA adsorption and made the system non-porous. The values of R^2^ and Q_e,exp_ also confirmed this hypothesis.

As for the evaluation of the binding properties of the developed systems for TBBPA using equilibrium adsorption experiments, the adsorption isotherms for the MIPs obtained from samples containing APTES and APTES-HC are reported in Figure 7. The imprinted and non-imprinted CS-based systems were not taken into consideration, both due to the poor extraction capacities found and the absence of cavities capable of promoting the selective extraction of TBBPA.

In general, all systems showed an increase in the amount of TBBPA bound as the initial concentration of the TBBPA solution increased, until reaching a saturation for a flame retardant concentration value of around 4 ppm (Figure 7A). However, only the sample containing Fe_3_O_4_@APTES-HC0.7 displayed the highest capture capacity. In general, the NIP systems, compared to imprinted analogs, showed lower binding affinity for TBBPA due to the absence of specific cavities (see Appendix A), confirming the data reported in the literature [32].

To investigate the interaction between the adsorbate and the sorbent, the equilibrium adsorption data were fitted with Langmuir and Freundlich equations. Figure 7B,C shows the fit of the isotherms of the Fe_3_O_4_@APTES-HC0.7-MIP/NIP sample, since this was the most promising system from the application point of view, while in Table 4 the values of the parameters obtained using the two models are reported.

It is known that the Langmuir equation has been applied in many monolayer adsorption processes, and the Freundlich equation is believed to be suitable to describe the adsorption behavior of non-covalent MIPs [68,69]. The reported data confirmed the better ability of Freundlich model to define the adsorption isotherm of the Fe_3_O_4_@APTES-HC0.7-MIP system, given a high value of *R*^2^. Furthermore, this system had a value of *n*^−1^ ranging from 0.27 to 0.41. This parameter confirmed the nonlinearity of the isotherms, which could be attributed to the heterogeneity of the adsorption sites, electrostatic attractions, and other sorbent–sorbate interactions present in the MIP system and not in the unimprinted one [70]. Finally, the *K_F_* value (adsorption capacity of a substance on a given surface at a given temperature) was higher for the MIP system, indicating the high affinity of this system for the capture of TBBPA.

These data agreed with the works reported in the literature [59,67]. Analyzing the Fe_3_O_4_@APTES-HC0.7-NIP system, it was observed that the Langmuir model was more suitable to define the adsorption isotherm. Indeed, in this system, the adsorption process depended on the polymeric monolayer formed during in situ polymerization and was not dependent on the presence of heterogeneous adsorption sites.

### 2.8. Selectivity of the Prepared MMIPs

To demonstrate the specificity of the Fe_3_O_4_@APTES-HC0.7-MIP/NIP systems in recognizing TBBPA, other flame retardants, 4BP and 2,4DBP (see Materials and Methods section), were introduced into the selective removal experiments. As shown in Figure 8, both materials were able to extract the pollutants selected in this study to some extent. However, the MIP system showed a much higher binding capacity towards TBBPA, as also evidenced by the selectivity values (ε) reported in Appendix A.

As expected, the Fe_3_O_4_@APTES-HC0.7-NIP system did not show high selectivity towards TBBPA. In fact, for both tested pollutants, as interferents in the adsorption process, an affinity value equal to about 1 or even lower than 1 could be observed. This behavior was due to the absence of specific cavities, which consequently led to an adsorption driven by weak and long-range interactions. Instead, the values obtained for the Fe_3_O_4_@APTES-HC0.7-MIP sample showed ε values higher than 1, indicating the high recognition of TBBPA due to the imprinted nature of the system. Although adsorption of the pollutants used as interferents by the MMIP system was observed, due to their similar structure to that of TBBPA, the extraction capacity of this matrix towards TBBPA was found to be better, allowing us to confirm the possible use of this system in real environmental applications aimed at capturing TBBPA.

The same behavior was also highlighted by Shao et al., who reported a better affinity for TBBPA of the MMIP system if compared with the non-imprinted system and a lower extraction capacity of the latter towards the analog pollutants, which in their case was around 10% [32].

### 2.9. Reusability of the Prepared MMIPs

To enable the application of these systems in the environmental field, the evaluation of the reusability of the adsorbent is of fundamental importance [32,71]. Figure 9 shows the adsorption efficiency (AE%) and the desorption efficiency (DE%) of the Fe_3_O_4_@APTES-HC0.7-MIP/NIP system, chosen based on its high extraction capacities. From the obtained results, it could be stated that, after 24 h of contact with the analyte, the non-imprinted samples still had good extraction capacities, which, however, were lower than the imprinted analogs, confirming non-specific absorption of the analyte. The good desorption percentages observed in the first cycle of use allowed these systems to recharge the analyte, also allowing them to reach extraction efficiency values for the Fe_3_O_4_@APTES-HC0.7-MIP system of around 77% in the second cycle and around 67% in the third cycle. Since the washing steps were not able to completely remove the analyte, the subsequent reuse cycle was affected by the presence of the residual TBBPA, reducing the performance of the MIP and NIP systems.

To verify the presence of any residual traces of the template that was not removed, SEM-EDX measurements were performed on the sample Fe_3_O_4_@APTES-HC0.7-MIP and on the same sample after washing with an acetonitrile–H_2_O mixture. Appendix A shows the EDX spectra of these systems (Appendix A, respectively). As can be seen, the bromine peak was still observable in spectrum B, demonstrating the presence of an unextracted template. By measuring the ratio between the atomic percentage of bromine and the atomic percentage of carbon, it was observed that this value decreased after washing but was still measurable (%Br/%C from 0.40 ± 0.10 to 0.10 ± 0.05). Probably, the traces of the template trapped in the underlying surface layers were due to good interaction established with the antioxidant molecules, which limited its backscattering.

Since SEM analysis ruled out possible morphological changes in the samples (Appendix A) and TGA measurements ruled out possible degradation of the polymer shell (same weight loss of the sample before desorption), complete removal of the template might be achieved either by using longer washing times or solvents capable of better penetrating the mass of the system.

## 3. Materials and Methods

### 3.1. Materials

Iron(III) chloride hexahydrate (FeCl_3_·(H_2_O)_6_) and iron(II) chloride tetrahydrate (FeCl_2_·(H_2_O)_4_), chitosan (CS, Mn = 280,000 g/mole, viscosity 200–800 cP at 25 °C) [72], (3-aminopropyl)triethoxysilane (APTES), hydrochloric acid (36.5–38.0%,), methacrylic acid (stabilized with hydroquinone monomethyl ether), 98% ethylene glycol dimethacrylate (EGDMA) containing 90–110 ppm monomethyl ether hydroquinone as an inhibitor, 98% 3,4-dihydroxyhydrocinnamic acid (HC), 2,2-diphenyl-1-picrylhydrazyl (DDPH), and ninhydrin (NHN) were purchased from Sigma Aldrich (Saint Louis, MO, USA) and used as received. The other chemicals were analytical-grade and purchased from Sigma-Aldrich.

### 3.2. Preparation of Magnetic Nanoparticles (MNPs)

Magnetite nanoparticles were synthesized using a method similar to the original method of Massart et al. [73], but maintaining the basic environment reported by Qu et al. [74]. In particular, a 2:1 molar ratio between the two iron salts was used. Initially, 2 M FeCl_3_·(H_2_O)_6_ was dissolved in 7.5 mL of 2 M HCl providing the required Fe^3+^ ions, while 1 M FeCl_2_·(H_2_O)_4_ was dissolved in 7.5 mL of distilled H_2_O, providing the Fe^2+^ ions. The two solutions were then combined to allow complex formation and quickly poured into 200 mL of 1 M NH_3_. The solution immediately turned a deep black color. The reaction was left under stirring for 30 min. Finally, the precipitate was separated from the supernatant using a neodymium magnet (1.52 T) (in this case, a clear and rapid separation of the particles from the solution could be observed). Once the supernatant was removed, the pH was neutralized with distilled water. Subsequently, the nanoparticles were washed with THF and allowed to air dry. Once the precipitate was sampleable, it was dried in an oven at ≈150 °C and then ground finely with an agate mortar and stored. The acronym used for the obtained MNPs was Fe_3_O_4_.

### 3.3. Preparation of CS-Coated MNPs

Initially, 40 mg of CS was dissolved in 2 mL of a 2% *v*/*v* acetic acid solution and left stirring overnight to allow complete dissolution of CS. Then, the resulting solution was placed in contact with the nanoparticles in a 1:5 ratio (40 mg of polymer per 200 mg of MNP) for 30 min. The reaction was carried out at room temperature in an ultrasonic bath. Next, the CS-coated MNPs were recovered with a neodymium magnet. Once the supernatant was removed, the MNPs were washed with 1 mL of 1 M NaOH and distilled H_2_O. Finally, MNPs were frozen and then lyophilized overnight. The acronym used for the coated MNPs was Fe_3_O_4_@CS.

### 3.4. Preparation of APTES-Coated MNPs

The introduction of APTES onto the surface of MNPs was performed following the procedure described by Erogul et al. [75]. Specifically, 200 mg of nanoparticles were suspended in 1 mL of distilled water, the pH of which was adjusted by adding 1% *v*/*v* acetic acid, thus promoting proper hydrolysis of silane. Then, 3 mL of APTES was added, a quantity modified from the original procedure as it was observed that this provided better coverage. The reaction was carried out in an ultrasonic bath at 40 °C for 3 h. At the end of the reaction, the preparation was separated from the overlying liquid with the help of a magnet, subjected to an initial wash with 1 mL of 1 M NaOH and further washes in a centrifuge with distilled water, and it was finally frozen and lyophilized overnight. The acronym used for the coated MNPs was Fe_3_O_4_@APTES.

### 3.5. Covalent Introduction of 3,4-Dihydroxyhydrocinnamic Acid (HC) on APTES-Coated MNPs

Once the amino groups were introduced on the surface of the Fe_3_O_4_@APTES sample, to promote the interaction of the matrix with the chosen template and modulate the thickness of the subsequent polymer shell, the carboxyl groups of the HC antioxidant were covalently bonded to the magnetic system using 3 different molar ratios between APTES and HC, 1:0.3; 1:0.7; 1:1, respectively. Specifically, 100 mg of Fe_3_O_4_@APTES nanoparticles suspended in 1 mL of distilled H_2_O were added dropwise to a solution, previously sonicated for 15 min, of HC, EDC and NHS, in a ratio of 1:2:4. The reaction was carried out at room temperature for 6 h under stirring. After removing the nanoparticles with a magnet, they were washed several times with water to eliminate any HC molecules not covalently bound to the MNPs. To verify the effectiveness of this treatment, the wash water was collected and subjected to UV measurements each time until no antiradical activity due to HC released from the systems was observed. The functionalized nanoparticles were then frozen and lyophilized. The acronyms used for the obtained systems were Fe_3_O_4_@APTES-HC0.3, Fe_3_O_4_@APTES-HC0.7, and Fe_3_O_4_@APTES-HC1.0.

### 3.6. Synthesis of Magnetic Molecularly Imprinted Polymer Systems (MMIPs)

Preparation of MMIPs was performed following the synthesis reported by Hemmati et al. with some modifications [76]. Specifically, 80 mg of each Fe_3_O_4_-based system previously coated with CS, APTES, or APTES and HC were placed in a three-neck flask containing 80 mL of acetonitrile. Before proceeding with the polymerization reaction, TBBPA (0.32 mmol) was introduced into the flask, which was then placed in an ultrasonic bath for 15 min. The sonication step was necessary to allow proper dispersion of the nanoparticles in the reaction medium. Then, 2 mmol of ethylene glycol dimethacrylate (EDGMA), used as a crosslinker, 80 mg of benzoyl peroxide (BPO) as a radical initiator, and 4 mmol of methacrylic acid (MAA) were added. The reaction was then carried out at 80 °C, under stirring, for 3 h in a nitrogen atmosphere. Once the polymerization process was completed, the nanoparticles were separated from the solvent with a magnet. The first wash was performed with acetone and then with ethanol. To remove any TBBPA molecules still present in the preparation, MMIPs were subjected to 3 additional washes of 1 h each, under stirring, using a solution (3 mL) of acetic acid–methanol in a 1:9 ratio. After removing any residual acetic acid, further washing the sample once with methanol and then with distilled water, it was frozen and lyophilized.

Non-imprinted analogs (MNIPs) obtained in the absence of TBBPA and by the same procedure as above were also synthesized. The acronyms used for the obtained MMIPs and MNIPs were Fe_3_O_4_@CS-NIP/MIP, Fe_3_O_4_@APTES-NIP/MIP, Fe_3_O_4_@APTES-HC0.3-NIP/MIP, Fe_3_O_4_@APTES-HC0.7-NIP/MIP, and Fe_3_O_4_@APTES-HC1.0-NIP/MIP.

### 3.7. Infrared Spectroscopy

The successful surface functionalization of MNPs and subsequent production of MMIPs was evaluated by Fourier transform infrared spectroscopy (FTIR). Spectra were acquired in attenuated total reflection (ATR) by a Nicolet 6700 (Thermo Fisher Scientific, Waltham, MA, USA) equipped with a Golden Gate single-reflection diamond ATR accessory at a resolution of 2 cm^−1^ and adding 200 scans.

### 3.8. Morphological Analysis

The morphology and dimensions of MNPs were studied by field emission scanning electron microscopy (FESEM, AURIGA Carl Zeiss AG, Oberkochen, Germany). The diameters of MNPs and MMIPs were determined using ImageJ software (version 1.53k). The images of each sample were analyzed, and the diameters were measured at 100 different positions, from which the mean value ± standard deviation was obtained for each sample.

Microanalysis was carried out with EDX (energy dispersive X-ray spectroscopy, Bruker Quantax, Berlin, Germany). EDX measurements were carried out both on the sample Fe_3_O_4_@APTES-HC0.7-MIP and on the same sample after washing with the acetonitrile–H_2_O mixture. The Br at.%/C at.% ratio was determined by the ratio of the atomic % of bromine with respect to the atomic % of carbon.

### 3.9. Determination of Introduced Amino Groups

The amino groups introduced on the surface of MNPs were determined by ninhydrin assay (NHN) [77]. Initially, a 2% (*w*/*v*) ninhydrin solution in 2-isopropanol was prepared. Then, 3 mg of modified MNPs were placed in 2 mL of ninhydrin solution and allowed to react at 90 °C for 15 min using a cooling system. During this time, the solution took on a deep purple color related to the reaction between ninhydrin and free amino groups. The solution was then allowed to cool for 20 min, after which the absorbance was measured at 570 nm using a UV-Vis spectrophotometer (HP DIODE ARRAY, Agilent Technologies Inc., Palo Alto, CA, USA). The concentration of NH_2_ groups normalized to the mg of sample analyzed was determined using calibration curves obtained by plotting the concentration of CS (mg/mL) and APTES (mg/mL) against absorbance. All the measurements were performed in triplicate, and the values were reported as the mean values ± standard deviation.

### 3.10. Determination of HC Antioxidant Content on MNPs by the DPPH Method

Evaluation of the amount of HC introduced into the nanosystems was carried out using 2,2-diphenyl-1-picryl-hydrazyl (DPPH) as a radical model, according to the method developed by Marsden Blois and later modified by Brand-Williams, Cuvelier, and Berset [78]. Specifically, 3 mg of Fe_3_O_4_@APTES-HCX (X = 0.3, 0.7, and 1.0) were put in contact with 2 mL of 0.2 mM DPPH solution and 2 mL of methanol for 1 h, the time necessary to reach plateau. At completion, the absorbance of the solutions was measured at a wavelength of 520 nm. The concentration of the antioxidant molecules per mg of the tested sample was obtained by means of a calibration curve obtained using a 0.2 mM DPPH solution in methanol and several aliquots of 1 × 10^−4^ M HC solution, allowing the reaction to take place for 1 h in the absence of light. Furthermore, through the DPPH test, it was possible to trace the antiradical capacity (*AC* %) of the systems, calculated as follows:$$AC\;\left(\%\right)=\;\frac{{Abs}_{0}-{Abs}_{s}}{{Abs}_{0}}\;\times \;100$$
where *Abs*_0_ represented the absorbance of the DPPH solution without MNPs and *Abs_s_* instead indicated the absorbance of the DPPH solution after 1 h contact with MNPs. All measurements were performed in triplicate and the values were reported as the mean values ± standard deviation.

### 3.11. Elemental Analysis

The chemical compositions of the functionalized MNP and MMIP systems was determined by a Carlo Erba Instrument EA 1110 CHNS-O elemental analyzer (Carlo Erba Reagents SAS, Chausséedu Vexin, France). The degree of functionalization of the systems containing CS and APTES was derived from the %N/C ratio. For the MMIP and MNIP systems, it was possible to trace the C content (%/mg) normalized to the mg of nanocomposite material used for the test.

### 3.12. Thermogravimetric Analysis

The thermal stability of the developed systems was evaluated using thermogravimetric analysis. This analysis made it possible to verify the degree of coverage of the MNPs and MMIPs and to evaluate their degradation temperature (Td). The analysis was conducted using a Mettler TG 50 thermobalance (Mettler Toledo, Columbus, OH, USA). For the measurements, a temperature ramp with a heating rate of 10 °C·min^−1^ was used under N_2_ flow in the temperature range 25–500 °C. The weight losses attributable to the degradation of the organic material were considered in this range for all samples.

### 3.13. Study of TBBPA Adsorption

To test the adsorption capacity of the prepared MIP and NIP systems, approximately 10 mg of each sample was brought into contact with 1.5 mL of TBBPA solution (0.5–6 ppm). A stock solution was prepared by dissolving a weighed amount of TBBPA in a 50:50 solution of acetonitrile and water. The flame-retardant solution was then diluted with the same mixture to the chosen concentration and the pH adjusted to 5 by adding HCl. Quantitative analysis of TBBPA adsorption was performed using a Kontron Instrument (Everett, MA, USA) HPLC PUMP 422 high-performance liquid chromatograph coupled with a UV-Vis detector (Jasco UV-2070, Easton, MD, USA). The system is equipped with a C-18 reversed-phase column (250 4.6 mm, particle size 5 μm). The injection volume and mobile phase flow rate were set to 20 μL and 1 mL/min, respectively. An isocratic mobile phase containing H_2_O/acetonitrile 5:95 was used, and the detector was set to a wavelength of 207 nm.

A calibration curve was created to enable quantification of TBBPA. The adsorption kinetics were monitored by taking 60 μL aliquots every 30 min for up to 3 h, at which point a plateau was reached, and then after 24 h to obtain the adsorbed concentration at equilibrium. The amount of TBBPA adsorbed per unit mass of magnetic adsorbent (*Qt*, mg/g) was calculated by the following equation:$$Q_{t}=\;\frac{(C_{0}-C_{t})\times V}{m}$$
where *C*_0_ is the initial concentration of TBBPA (mg/L), *C_t_* is the concentration of TBBPA at different times, m is the mass expressed in grams of Fe_3_O_4_@MIP or Fe_3_O_4_@NIP, and *V* is the volume of the aqueous solution. When the sample reached the adsorption plateau, the *Q_e_* value was obtained. The adsorption efficiency percentage (*AE*%) was expressed as follows:$$AE\;\left(\%\right)=\left(\frac{Amount\;of\;adsorbed\;TBBPA\;(mg)}{Initial\;pesticide\;amount\;(mg)}\right)\times 100$$

Three desorption steps were performed using 1.5 mL of acetonitrile–water (50:50) for 24 h. The amount of TBBPA desorbed was calculated for each fraction, and the desorption efficiency percentages (*DE*) were calculated as follows:$$DE\;\left(\%\right)=\left(\frac{Amount\;of\;desorbed\;TBBPA\;(mg)}{Initial\;pesticide\;amount\;(mg)}\right)\times 100$$

Whereas, the affinity value (α) was obtained using the following equation:$$\alpha =\frac{Q_{MMIP}}{Q_{MNIP}}$$
where *Q_MMIP_* and *Q_MNIP_* were the binding capacities, expressed as the amount of TBBPA adsorbed at each time point per g of adsorbent phase, for the MMIP and MNIP systems, respectively.

In addition, to examining the kinetic mechanism, pseudo-first-order (1) and pseudo-second-order (2) models were employed using the following equations:(1)$$\ln\left(Q_{e}-Q_{t}\right)=\ln Q_{e}-K_{1}t$$(2)$$\frac{t}{Q_{t}}=\frac{1}{K_{2}{Q_{e}}^{2}}+\frac{1}{Q_{e}}t$$
where *Q_e_* is the adsorption capacity at equilibrium (mg/g), *Q_t_* is the adsorption capacity at time *t* (mg/g), t = time (min), and *K*_1_ (1/min) and *K*_2_ (g·mg^−1^·min^−1^) are the first order and second order rate constants, respectively.

As regards the isotherm adsorption, the models of Langmuir (3) and Freundlich (4) were used, and the data were fit by the following equations:(3)$$\frac{C_{e}}{Q_{e}}=\frac{1}{Q_{m}K_{L}}+\frac{Ce}{Q_{m}}$$(4)$$\ln Qe=\ln K_{F}+\frac{1}{n}\ln Ce$$
where *C_e_* is the equilibrium concentration of TBBPA, *Q_e_* is the equilibrium binding capacity for MMIPs, *Q_m_* is the maximum binding capacity, and *K_F_* and *K_L_* are model constants related to binding capacity and binding energy (*L* mg^−1^). All the parameters were predicted from the plot between ln*Q_e_* and ln*C_e_* and between *C_e_*/*Q_e_* and *C_e_*, respectively, for the two models. All adsorption and desorption measurements were performed in triplicate, and data are reported as mean value ± standard deviation.

### 3.14. Selectivity of the Prepared MMIPs

To study the selectivity of the systems towards TBBPA capture, the most promising samples were put in contact with solutions containing TBBPA and some structural analogs such as 4-bromophenol (4BP) and 2,4-dibromophenol (2,4DBP), whose structures are shown in Figure 10. Specifically, solutions of TBBPA and the two analogs 4BP and 2,4DBP were prepared at the same concentration (4 ppm) and put in contact with 10 mg of MMIPs or MNIPs. Subsequently, under stirring for 24 h, the imprinted and non-imprinted systems were separated from the solution by the use of a magnet. The amount of TBBPA or analogs in the supernatant was detected by HPLC, using the calibration curves obtained at 292 nm and 262 nm for TBBPA and analogs, respectively. Next, the removal efficiency and the relative adsorption capacity were calculated. The latter allowed us to determine the selectivity value (ε), calculated as follows:$$\varepsilon =\frac{Q_{TBBPA}}{Q_{4Br-Ph}};\;\varepsilon =\frac{Q_{TBBPA}}{Q_{2,4dBr-Ph}}$$
where *Q_TBBPA_*, *Q*_4*BP*_, and *Q*_2,4*DBP*_ represent the adsorption capacities of TBBPA, 4BP, and 2,4DBP by the system under analysis, respectively. All experiments were performed in triplicate.

### 3.15. Reusability of MMIPs

The most promising MIP and NIP systems were tested for reuse by employing a TBBPA solution (1.5 mL) with a concentration of 4 ppm for the adsorption tests and a solution of acetonitrile and water, in a 1:1 ratio, for the flame retardant discharge step. Briefly, the solution was placed in contact with the sample (10 mg) for 24 h until maximum adsorption was reached. Then, at the end of each analysis, the sample was washed with 1.5 mL of acetonitrile–water (50:50) three times, dried, and put back in contact with a new TBBPA solution. The process was repeated a total of 3 times.

### 3.16. Statistics

Analysis of variance comparisons were performed using Mini-Tab. Differences were considered significant for *p* < 0.05. Data are reported as means ± SD.

## 4. Conclusions

In this work, new MIP systems were developed based on differently functionalized magnetite nanoparticles as adsorbent materials for capturing TBBPA, a known flame retardant and persistent pollutant. To stabilize the magnetic nanosystems and form new binding sites for further functionalization, the MNPs were coated either with a low-molecular-weight molecule (APTES) or a polymer (CS). In addition, to introduce appropriate functional groups able to establish greater interactions with the chosen analyte and to control the thickness of polymer shell subsequently formed by the preparation of MIPs, the APTES nanocomposites were further functionalized with 3,4-dihydroxyhydrocinnamic acid (HC) using different APTES:HC molar ratios. To obtain MIPs, the prepared nanocomposites were then subjected to in situ radical polymerization using methacrylic acid (MAA) as the monomer, ethylene glycol dimethacrylate (EDGMA) as the crosslinker and TBBPA as the template. To verify the performance of the imprinted matrices, systems obtained in the absence of the template (MNIP) were also prepared. Infrared, thermal, and elemental analyses were used to confirm the functionalization and to quantify the degree of coating of the nanocomposites (14–24%). The ninhydrin test, on the other hand, allowed quantification of the number of introduced amino groups and, particularly in the case of the Fe_3_O_4_@APTES systems, subsequent functionalization with varying amounts of antioxidant (APTES:HC molar ratios of 1:0.3, 1:0.7, and 1:1). The DPPH assay was useful to verify the surface functionalization with the antioxidant molecule and confirm the increase in the antiradical capacity (AC%) of the systems as the HC concentration increased (AC from 34 to 66%). The subsequent in situ polymerization step required to produce MMIPs and MNIPs was verified qualitatively by FT-IR spectroscopy and evaluated quantitatively via elemental analysis. These analyses showed a higher coating for Fe_3_O_4_@CS-MIP/NIP systems than for those containing APTES. The in situ polymerization process maintained the spherical morphology of the nanosystems, evaluated by FESEM, while increasing their diameter, particularly in the case of the CS-containing system (d = 77 nm), confirming the higher degree of coating. The extraction capacity of the nanosystems, assessed by high-performance liquid chromatography (HPLC-UV), evidenced the good performance of the MMIP systems when compared to the non-imprinted analogs. However, when comparing the Fe_3_O_4_@CS-MIP and Fe_3_O_4_@APTES-MIP systems, it was possible to observe the higher extraction capacity of the latter system, equal to 50%, given the reduced size of the polymeric shell and the consequent increase in the surface-to-volume ratio. As for the systems containing the antioxidant, it was noted that the Fe_3_O_4_@APTES-HC0.7-MIP system showed the best extraction efficiency (AE = 85%), which was also associated with a good discharge efficiency (DE = 70%). Analysis of the adsorption kinetic data evidenced pseudo-second-order kinetics for this system, confirming the presence of specific interactions between the support and analyte that drove the adsorption process. The adsorption isotherms confirmed this hypothesis, as the Freundlich model was ideal for defining the behavior of the system Fe_3_O_4_@APTES-HC0.7-MIP. For the other two HC-containing systems, however, no improvement in MMIP extraction capacity could be found. This was attributed to the active contribution of the antioxidant in radical capture during the polymerization of MMA, which reduced the polymer layer during MIP formation and, thus, the production of specific cavities. Selectivity measurements of the Fe_3_O_4_@APTES-HC0.7-MIP system, towards mixtures of flame retardants formed by TBBPA and 4-bromophenol (4BP) and TBBPA and 2,4-dibromophenol (2,4DBP), showed a value of ε > 1, highlighting the potential of this matrix in the treatment of real environmental matrices. Finally, reuse experiments confirmed good extraction efficiency for the Fe_3_O_4_@APTES-HC0.7-MIP system, with values of 77 and 67% found in the second and third cycles, respectively.

## Figures and Tables

**Figure 1 ijms-26-07686-f001:**
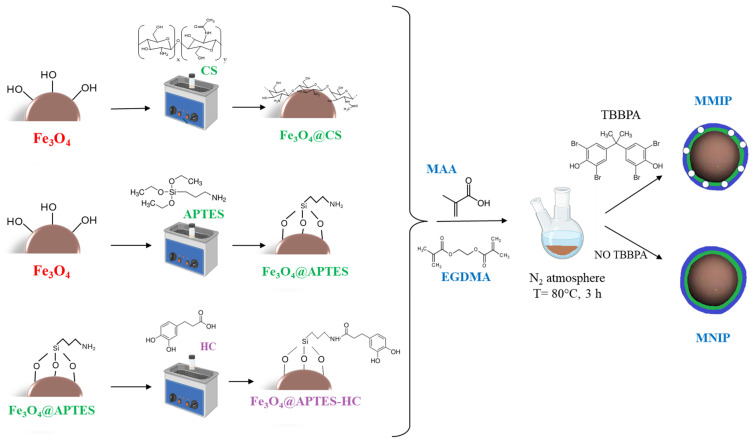
MMIP and MNIP system preparation scheme.

**Figure 2 ijms-26-07686-f002:**
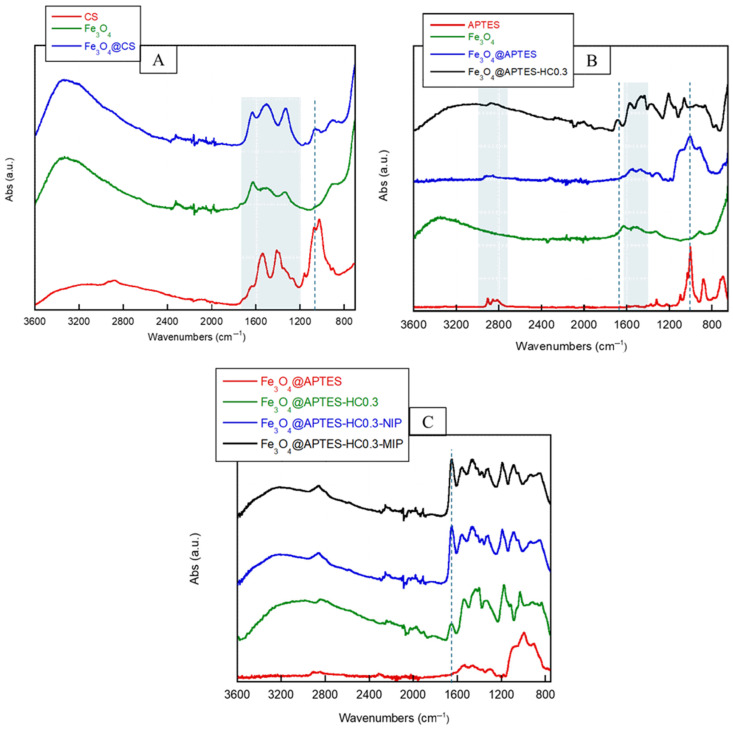
FTIR spectra of pristine chitosan (CS); pristine magnetite NPs (Fe_3_O_4_) and chitosan-coated MNPs (Fe_3_O_4_@CS) (**A**); FTIR spectra of pristine APTES, pristine magnetite NPs (Fe_3_O_4_), APTES-coated MNPs (Fe_3_O_4_@APTES), and NPs functionalized with APTES and then HC (molar ratio APTES:HC 1:0.3) (Fe_3_O_4_@APTES-HC0.3) (**B**); FTIR spectra of APTES-coated MNPs (Fe_3_O_4_@APTES), Fe_3_O_4_@APTES NPs functionalized with HC (molar ratio APTES:HC 1:0.3) (Fe_3_O_4_@APTES-HC0.3), and MMIP and MNIP systems obtained by in situ polymerization of MAA and EGDMA on Fe_3_O_4_@APTES-HC0.3 NPs (**C**).

**Figure 3 ijms-26-07686-f003:**
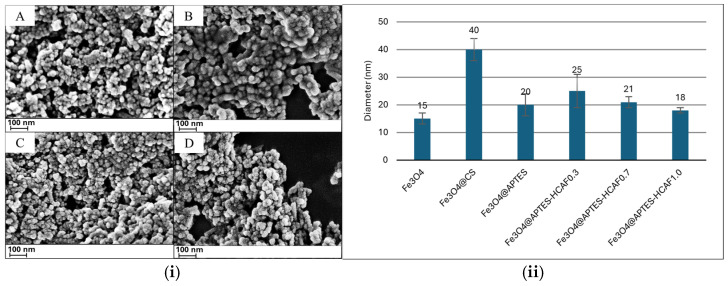
(**i**) SEM micrographs of obtained MNPs: Fe_3_O_4_ (**A**); Fe_3_O_4_@CS (**B**); Fe_3_O_4_@APTES (**C**); Fe_3_O_4_@APTES-HC0.3 (**D**). (**ii**) Diameter of prepared systems.

**Figure 4 ijms-26-07686-f004:**
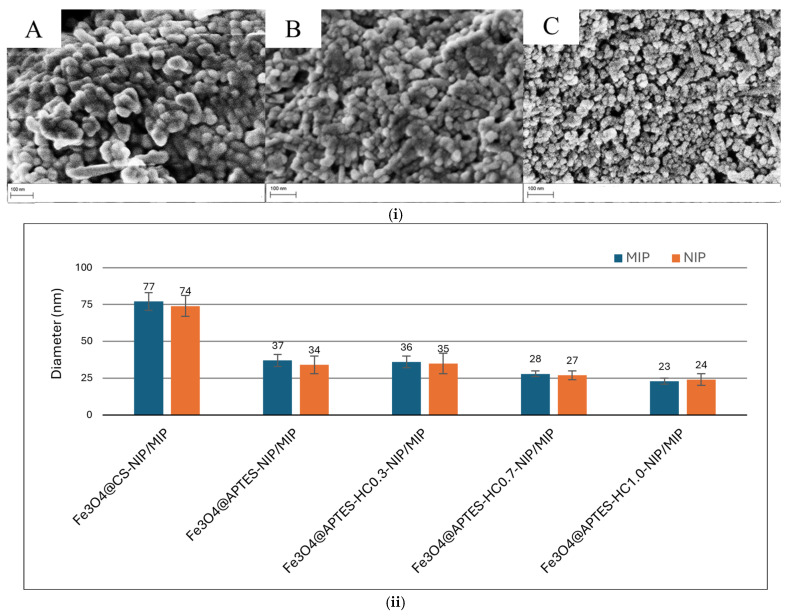
(**i**) SEM micrographs of the obtained MMIP and MNIP systems: MMIP-CS (**A**); MNIP-APTES (**B**); MMIP-APTES-HC1.0 (**C**). (**ii**) Diameter of prepared systems.

**Figure 5 ijms-26-07686-f005:**
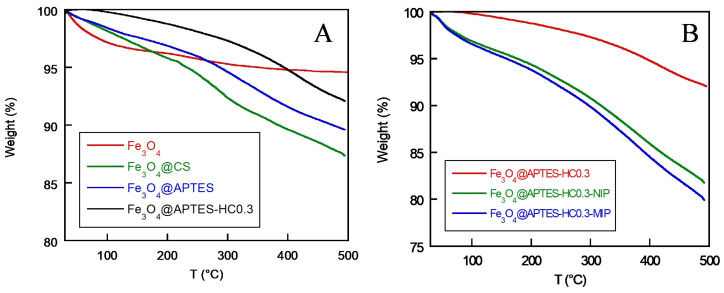
Thermogravimetric curves of naked magnetite NPs (Fe_3_O_4_), chitosan-coated MNPs (Fe_3_O_4_@CS), APTES-coated MNPs (Fe_3_O_4_@APTES), and APTES-coated and then HC-functionalized MNPs with an APTES:HC molar ratio of 1:0.3 (Fe_3_O_4_@APTES-HC0.3) (**A**); thermogravimetric curves of MMIP and MNIP systems obtained from the sample Fe_3_O_4_@APTES-HC0.3 (**B**).

**Figure 6 ijms-26-07686-f006:**
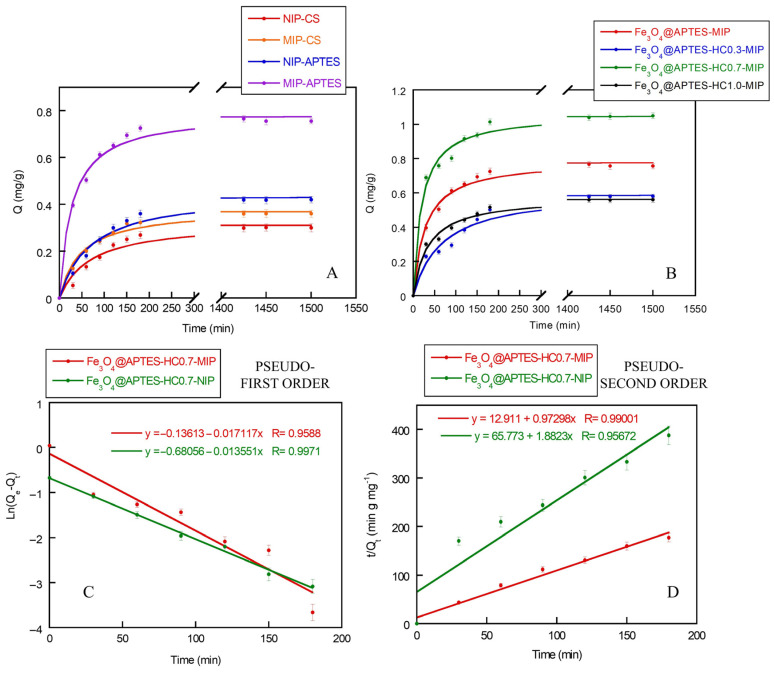
Adsorption kinetics of TBBPA on Fe_3_O_4_@CS-MIP/NIP and Fe_3_O_4_@APTES-MIP/NIP (**A**); adsorption kinetics of Fe_3_O_4_@APTES-MIP, Fe_3_O_4_@APTES-HC0.3-MIP, Fe_3_O_4_@APTES-HC0.7-MIP, and Fe_3_O_4_@APTES-HC1.0-MIP (**B**); Pseudo-first- (**C**) and pseudo-second- (**D**) order kinetics models for Fe_3_O_4_@APTES-HC0.7-MIP/NIP systems.

**Figure 7 ijms-26-07686-f007:**
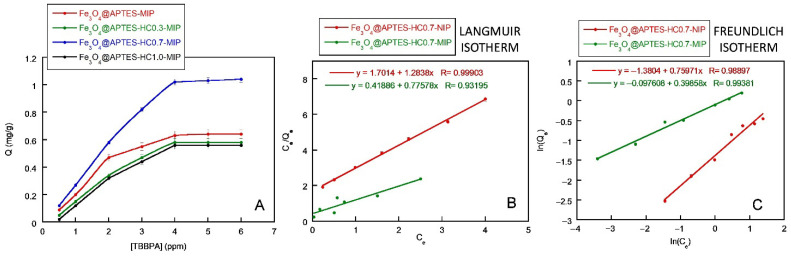
Adsorption isotherms of Fe_3_O_4_@APTES-MIP, Fe_3_O_4_@APTES-HC0.3-MIP, Fe_3_O_4_@APTES-HC0.7-MIP, and Fe_3_O_4_@APTES-HC1.0-MIP (**A**); Langmuir (**B**) and Freundlich (**C**) isotherms models for Fe_3_O_4_@APTES-HC0.7-MIP/NIP systems.

**Figure 8 ijms-26-07686-f008:**
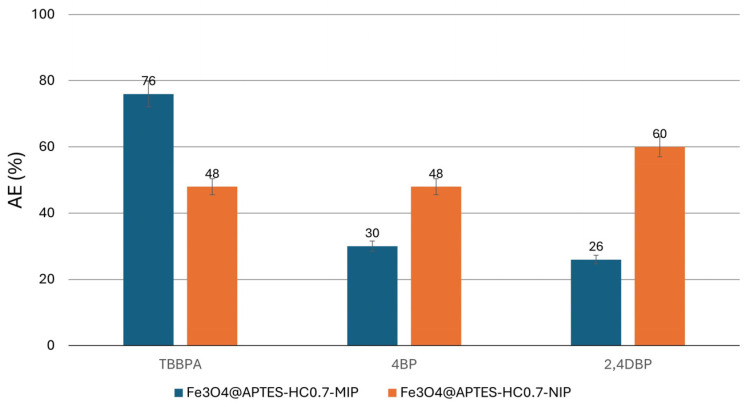
Binding selectivity of Fe_3_O_4_@APTES-HC0.7-NIP and Fe_3_O_4_@APTES-HC0.7-MIP systems for TBBPA, 4BP, and 2,4DBP.

**Figure 9 ijms-26-07686-f009:**
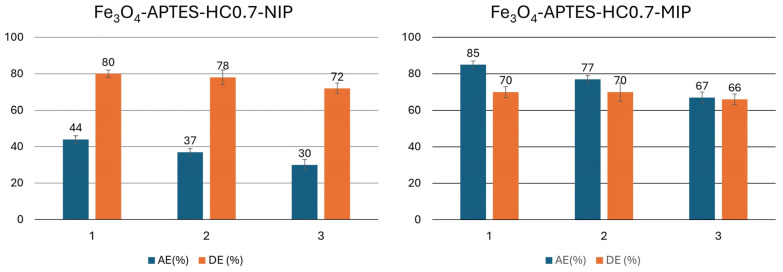
Adsorption efficiency (AE %) and desorption efficiency (DE %) of MNIP-APTES-HC0.7 and MMIP-APTES-HC0.7 systems.

**Figure 10 ijms-26-07686-f010:**
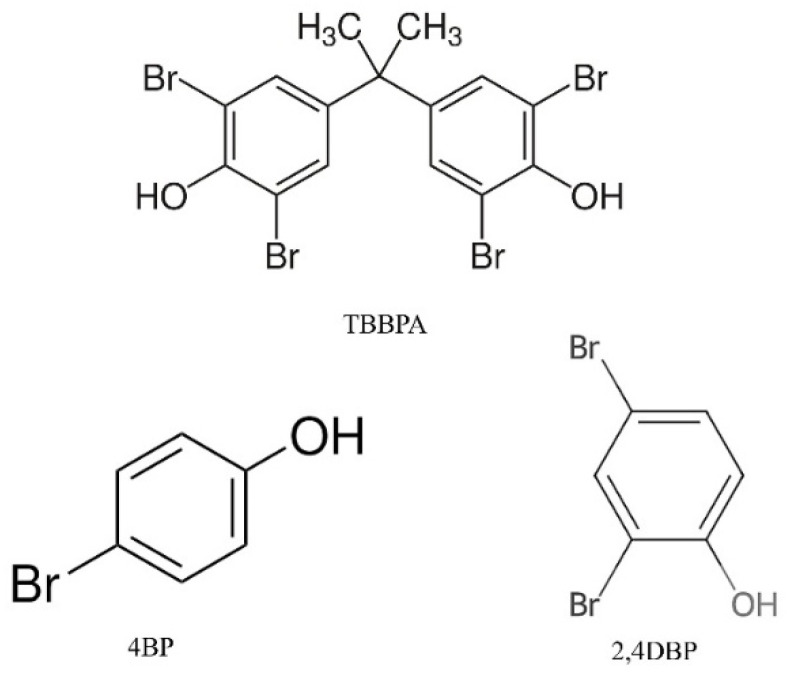
Structures of TBBPA, 4BP, and 2,4DBP.

**Table 1 ijms-26-07686-t001:** Theoretical and experimental % N/C.

Sample	Theoretical %N/C	Experimental %N/C
Fe_3_O_4_@CS	0.36	0.34 ± 0.03
Fe_3_O_4_@APTES	0.28	0.27 ± 0.04
Fe_3_O_4_@APTES-HC0.3	0.20	0.18 ± 0.02
Fe_3_O_4_@APTES-HC0.7	0.13	0.15 ± 0.02
Fe_3_O_4_@APTES-HC1.0	0.08	0.09 ± 0.03

**Table 2 ijms-26-07686-t002:** Percentage C content per mg of tested matrix.

Sample	C Content (%/mg)	Sample	C Content % (%/mg)
Fe_3_O_4_@CS-NIP	10.5 ± 0.3	Fe_3_O_4_@CS-MIP	10.4 ± 0.7
Fe_3_O_4_@APTES-NIP	6.5 ± 0.4	Fe_3_O_4_@APTES-MIP	6.3 ± 0.4
Fe_3_O_4_@APTES-HC0.3-NIP	6.0 ± 0.5	Fe_3_O_4_@APTES-HC0.3-MIP	6.0 ± 0.2
Fe_3_O_4_@APTES-HC0.7-NIP	3.1 ± 0.6	Fe_3_O_4_@APTES-HC0.7-MIP	3.1 ± 0.3
Fe_3_O_4_@APTES-HC1.0-NIP	1.0 ± 0.4	Fe_3_O_4_@APTES-HC1.0-MIP	0.9 ± 0.2

**Table 3 ijms-26-07686-t003:** Weight loss (%) and degradation temperature (T_d_) of the developed systems.

Sample	Weight Loss (%)	T_d_ (°C)	Sample	Weight Loss (%)	T_d_ (°C)
Fe_3_O_4_	4 ± 1	-	Fe_3_O_4_@APTES-HC0.3	6 ± 1	400 ± 5
Fe_3_O_4_@CS	13 ± 2	300 ± 4	Fe_3_O_4_@APTES-HC0.7	5 ± 1	410 ± 4
Fe_3_O_4_@APTES	9 ± 1	321 ± 3	Fe_3_O_4_@APTES-HC1.0	6 ± 2	420 ± 2
Fe_3_O_4_@CS-NIP	23 ± 3	376 ± 4	Fe_3_O_4_@CS-MIP	24 ± 2	370 ± 3
Fe_3_O_4_@APTES-NIP	18 ± 3	396 ± 5	Fe_3_O_4_@APTES-MIP	19 ± 3	393 ± 3
Fe_3_O_4_@APTES-HC0.3-NIP	18 ± 2	397 ± 5	Fe_3_O_4_@APTES-HC0.3-MIP	19 ± 1	395 ± 3
Fe_3_O_4_@APTES-HC0.7-NIP	16 ± 3	407 ± 4	Fe_3_O_4_@APTES-HC0.7-MIP	15 ± 2	404 ± 3
Fe_3_O_4_@APTES-HC1.0-NIP	14 ± 1	412 ± 3	Fe_3_O_4_@APTES-HC1.0-MIP	14 ± 1	410 ± 2

**Table 4 ijms-26-07686-t004:** Isotherms and kinetics parameters at 298 K for Fe_3_O_4_@APTES-HC0.7-MIP/NIP.

	Model	Parameters	Fe_3_O_4_@APTES-HC0.7-NIP	Fe_3_O_4_@APTES-HC0.7-MIP
Kinetics models	Pseudo-first order	*K*_1_ (min^−1^)	6.6 × 10^−5^	1.3 × 10^−4^
		*Q_e,cal_* (mg g^−1^)	0.56	0.90
		*Q_e,exp_* (mg g^−1^)	0.53 ± 0.01	1.04 ± 0.03
		R^2^	0.9890	0.9192
	Pseudo-second order	*K*_2_ (g·mg^−1^·min^−1^)	0.046	0.059
		*Q_e,cal_* (mg g^−1^)	0.56	0.82
		*Q_e,exp_* (mg g^−1^)	0.53 ± 0.01	1.04 ± 0.03
		R^2^	0.9153	0.9801
Isotherm models	Langmuir	*Q_m_* (mg g^−1^)	0.77	1.30
		*K_L_* (L mg^−1^)	0.45	3.07
		*R* ^2^	0.9981	0.8685
	Freundlich	*K_F_* (mg g^−1^)	0.25	0.91
		*n* (L mg^−1^)	1.32	2.5
		*R* ^2^	0.9781	0.9877

## Data Availability

The data necessary to reproduce the findings reported in this work can be obtained from the corresponding authors upon justified request.

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
