# Peer review of "Synthesis and Characterization of Magnetic Molecularly Imprinted Polymer Sorbents (Fe3O4@MIPs) for Removal of Tetrabromobisphenol A"

_ijms, 2025, doi:10.3390/ijms26167686_

Round 1
Reviewer 1 Report
Comments and Suggestions for Authors
This manuscript presents the development of molecularly imprinted polymer-coated magnetic nanoparticles for the selective capture of tetrabromobisphenol A. The study is systematic and potentially valuable. However, the following concerns should be addressed before the manuscript can be considered for publication:
- Lines 114-124, the authors summarize several adsorbents that have already demonstrated high selectivity for TBBPA. However, at the beginning of the following paragraph, they state that "producing a surface coating that is selective for capturing specific molecules within complex matrices is a major environmental challenge." This statement appears contradictory. Please clarify or reframe this discussion to resolve the apparent inconsistency.
- The authors claim that “to the best of our knowledge, there are no studies in the literature regarding the fabrication of engineered coatings able to specifically interact with the analyte to be captured.” This assertion is inaccurate. For instance, Shenashen et al. reported the development of mesoporous core/multi-shell silica nanoparticles for selective recognition and removal of toxic metals (Journal of Hazardous Materials 260 (2013): 833–843). Other similar examples also exist. Please revise this claim.
- Please avoid repeating background information already provided in the Introduction within the Results and Discussion section. This will help maintain the focus and clarity of the manuscript.
- In Figure 1, the alignment of the sub-figures should be improved for better visual presentation.
- Section 2.6, it is noted that none of the coated Fe₃Oâ‚„ samples reach a thermal plateau at 500 °C. Please explain how the weight loss values presented in Table 3 were derived.
- Please define 2,4DC in Line 479?
- For the reusability study, it is important to ensure that residual analytes from the previous cycle are completely removed before starting the next extraction cycle. Otherwise, the reusability performance may not reflect the intrinsic recovery behavior of the material.
- Did the authors evaluate the stability of the MIP-coated materials over time? For example, was there any observed change in extraction efficiency after storage? Including such data would strengthen the manuscript.
Author Response
REVIEWER 1
This manuscript presents the development of molecularly imprinted polymer-coated magnetic nanoparticles for the selective capture of tetrabromobisphenol A. The study is systematic and potentially valuable. However, the following concerns should be addressed before the manuscript can be considered for publication:
- Lines 114-124, the authors summarize several adsorbents that have already demonstrated high selectivity for TBBPA. However, at the beginning of the following paragraph, they state that "producing a surface coating that is selective for capturing specific molecules within complex matrices is a major environmental challenge." This statement appears contradictory. Please clarify or reframe this discussion to resolve the apparent inconsistency.
As suggested by the Reviewer, the statement has been rephrased (lines 126-128):
“However, the production of a selective surface coating capable of capturing specific molecules within complex matrices and with high performance still represents a major environmental challenge”.
- The authors claim that “to the best of our knowledge, there are no studies in the literature regarding the fabrication of engineered coatings able to specifically interact with the analyte to be captured.” This assertion is inaccurate. For instance, Shenashen et al. reported the development of mesoporous core/multi-shell silica nanoparticles for selective recognition and removal of toxic metals (Journal of Hazardous Materials 260 (2013): 833–843). Other similar examples also exist. Please revise this claim.
This sentence has also been reworded and some references has been added (lines 128-130):
“To the best of our knowledge, there are few studies in the literature on the fabrication of engineered coatings able to specifically interact with the analyte to be captured [33,34]”.
- Please avoid repeating background information already provided in the Introduction within the Results and Discussion section. This will help maintain the focus and clarity of the manuscript.
As suggested by the Reviewer, several information already present in the Introduction section have been removed from the Results and Discussion section.
- In Figure 1, the alignment of the sub-figures should be improved for better visual presentation.
We thank the reviewer for the suggestion. The alignment of the sub-figures has been carried out.
- Section 2.6, it is noted that none of the coated Fe₃Oâ‚„ samples reach a thermal plateau at 500 °C. Please explain how the weight loss values presented in Table 3 were derived.
Naked magnetite is stable up to temperatures of about 800°C without phase changes. Previous studies on magnetite coated with polymers or small molecules have shown that the organic material degrades in the range of 200–450°C. For this reason, the range explored was between room temperature and 500°C with weight loss attributable to the degradation of the organic material in the whole range for all samples.
- Please define 2,4DC in Line 479?
We apologize for the oversight. The acronym has been made explicit.
- For the reusability study, it is important to ensure that residual analytes from the previous cycle are completely removed before starting the next extraction cycle. Otherwise, the reusability performance may not reflect the intrinsic recovery behavior of the material.
We agree with the Reviewer on the need to completely remove the template before reusing the systems, otherwise system performance will be affected. However, to desorb the template the Fe3O4@APTES-HC0.7-MIP and Fe3O4@APTES-HC0.7-NIP samples were washed several times with the mixture acetonitrile:H2O (50:50). Unfortunately, as reported in the manuscript, the template remained entrapped in the organic coating as confirmed by new SEM-EDX measurements performed on the Fe3O4@APTES-HC0.7-MIP after desorption process (see Figure S4). In fact, EDX analysis evidenced the presence of bromine even after the numerous washings performed on the sample, demonstrating the difficulty of removing TBBPA. Probably, the flame retardant that penetrated deeper into the system had more difficulty backscattering after the formation of the polymer shell (MIP formation) due to strong interactions with HC (aromatic bond interactions). Since SEM analysis ruled out possible morphological changes in the samples (see Figure S4 A and B) and TGA measurements ruled out possible degradation of the polymer shell (data not shown), complete removal of the template might be achieved either by using longer washing times or solvents capable of better penetrating the mass of the system.
The newly data obtained from this analysis and the related discussion are reported in section Results and discussion, subsection 2.10 (lines 639-653):
“To verify the presence of any residual traces of the template that was not removed, SEM-EDX measurements were performed on the sample Fe3O4@APTES-HC0.7-MIP and on the same sample after washing with the acetonitrile:H2O mixture. Figure S4 shows the EDX spectra of these systems (Figure S4 A and B, respectively). As can be seen, the bromine peak was still observable in spectrum B, demonstrating the presence of an unextracted template. By measuring the ratio between the atomic percentage of bromine and the atomic percentage of carbon, it was observed that this value decreased after washings but was still measurable (%Br/%C from 0.40±0.10 to 0.10±0.05). Probably, the traces of the template trapped in the underlying surface layers were due to good interaction established with the antioxidant molecules, which limited its backscattering Since SEM analysis ruled out possible morphological changes in the samples (Figure S4) and TGA measurements ruled out possible degradation of the polymer shell (same weight loss of the sample before desorption), complete removal of the template might be achieved either by using longer washing times or solvents capable of better penetrating the mass of the system.”
- Did the authors evaluate the stability of the MIP-coated materials over time? For example, was there any observed change in extraction efficiency after storage? Including such data would strengthen the manuscript.
Long-term stability studies of the MIP-coated materials have not been conducted. Only experiments concerning the stability of the suspensions of the samples of interest (Fe3O4@APTES-HCX MIP samples) up to 48 hours was performed. This is because both the experiments to determine absorption efficiency (AE) and those to test the possible reuse of the systems were carried out over a 24-hour period. No variation in adsorption efficiency (AE%) was observed for the Fe3O4@APTES-HC0.3 system, while only a slight reduction was found for the Fe3O4@APTES-HC0.7 systems (AE=80%). In contrast, a greater reduction in AE (38%) was observed for the Fe3O4@APTES-HC1.0 system, likely attributable to a reduced surface/volume ratio due to partial aggregation phenomena of the NPs, as reported in the Results and Discussion section (subsection 2.7, lines 514-523) of the manuscript. Furthermore, images of the stability of some Fe3O4@APTES-HCX-MIP samples are shown in the supplementary information compared to that of the Fe3O4@APTES-MIP sample (Figure S2), demonstrating the influence of the antioxidant on solution dispersibility.
Lines 514-523:
“These results could also be attributed to the better dispersibility of the nanoparticles in the Fe3O4@APTES-HC0.7-MIP sample compared to that of the other two systems. Indeed, this system remained stable for at least 24 hours, showing only a slight aggregation of the nanoparticles at 48 hours (see Figure S2 A and B). With increasing HC concentration, aggregation phenomena were already visible after 24 hours (Figure S2 C). Although the adsorption kinetics were performed while keeping the samples under stirring, the reduced surface/volume ratio due to MNP aggregation could affect the availability of the specific cavities for TBBPA. In fact, although a small variation in adsorption efficiency was observed for the Fe3O4@APTES-HC0.7 systems at 48h (AE=80%), a greater reduction was evidenced for the Fe3O4@APTES-HC1.0 system (AE=38%).”

Reviewer 2 Report
Comments and Suggestions for Authors
This manuscript, reported by Clarissa Ciarlantini et al., reports the development of Fe₃O₄-based magnetic molecularly imprinted polymers (MMIPs). The study aims to address the challenge of selectively extracting the environmental contaminant TBBPA from complex media using reusable magnetic adsorbents.
While the study demonstrates that the HC-modified APTES-MMIPs exhibit promising extraction efficiency and reusability, there are several critical weaknesses in experimental design, data interpretation, and mechanistic evidence that must be addressed before the manuscript can be considered for publication. In particular, some of the assumptions regarding material structure and function lack sufficient experimental validation.
I recommend a major revision. A decision should be made after the authors have adequately responded to the following concerns:
1.The author claimed that CS was physically deposited (or through weak molecular interaction) while APTES was covalently bonded. However, given that chitosan can be chemically immobilized via aldehyde- or carboxyl-reactive crosslinkers (e.g., glutaraldehyde, EDC/NHS), the current comparison may be biased by differences in surface bonding stability and shell uniformity rather than by the intrinsic properties of the two materials. The authors should either justify this choice or prepare a CS-grafted control.
- The role of the antioxidant molecule HC in modulating polymer structure and performance is central to the study, but not mechanistically supported. The authors should explain why HC was selected over other radical scavengers, and whether the effect is unique to HC. Additional controls with other redox-active molecules are recommended.
3.The washing and purification steps are critical to remove physically adsorbed (non-grafted) species. However, no convincing evidence is provided that the materials were thoroughly purified. Without this, the observed HC-related effects might be due to unbound HC trapped in the polymer.
- FTIR spectra do not show convincing evidence of chemical interactions (e.g., no peak shifts or new bands indicating binding). Higher-resolution FTIR or 2D correlation spectroscopy could help resolve this.
- It is highly recommended to revise the abstract for the good readability. The current structure lacks a clear logical flow, the description is overly dense, and jumps into experimental details too quickly. It should be restructured to emphasize the research problem, proposed solution, key findings, and significance in a coherent order.
- The study does not discuss the colloidal stability or dispersibility of the nanoparticles in aqueous solution. Given that dispersibility affects adsorption efficiency and magnetic recovery, the authors should provide at least one of the following characterizations: the optical image of the dispersion (e.g., to show the Tyndall effect), DLS, zeta potential, or sedimentation data.
- The authors attribute the performance decline over reusability cycles solely to incomplete template desorption. Other factors, such as cavity collapse, matrix aging, or shell degradation, are not considered. The authors should characterize reused particles (e.g., by SEM/FTIR) to verify material stability.
Author Response
REVIEWER 2
This manuscript, reported by Clarissa Ciarlantini et al., reports the development of Fe₃O₄-based magnetic molecularly imprinted polymers (MMIPs). The study aims to address the challenge of selectively extracting the environmental contaminant TBBPA from complex media using reusable magnetic adsorbents.
While the study demonstrates that the HC-modified APTES-MMIPs exhibit promising extraction efficiency and reusability, there are several critical weaknesses in experimental design, data interpretation, and mechanistic evidence that must be addressed before the manuscript can be considered for publication. In particular, some of the assumptions regarding material structure and function lack sufficient experimental validation.
I recommend a major revision. A decision should be made after the authors have adequately responded to the following concerns:
- The author claimed that CS was physically deposited (or through weak molecular interaction) while APTES was covalently bonded. However, given that chitosan can be chemically immobilized via aldehyde- or carboxyl-reactive crosslinkers (e.g., glutaraldehyde, EDC/NHS), the current comparison may be biased by differences in surface bonding stability and shell uniformity rather than by the intrinsic properties of the two materials. The authors should either justify this choice or prepare a CS-grafted control.
The reasons for choosing CS are given in the manuscript (lines 139-143).
“In the case of the polymer coating, a polysaccharide such as CS was chosen, which contain many amino and hydroxyl groups that could be exploited both for its physical immobilization on the surface of magnetite NPs and to promote interaction with the target analyte during the formation of MIPs. In addition, CS is known for its high adsorptive properties, which are widely exploited in the analytical field. “
We did not want to use chemical reagents to chemically bind the polysaccharide, either in its original or modified form (carboxymethyl chitosan), to the surface of the MNPs, in order to have as many functional groups as possible available for subsequent interactions with the model. Furthermore, the stability of the coating was ensured by treating the MNPs with 1 M NaOH after a brief absorption of the polysaccharide (30 min), as reported in the materials and methods section (3.3 section). In fact, the restoration of the amino groups transforms the CS into a material that is completely insoluble in an aqueous environment.
- The role of the antioxidant molecule HC in modulating polymer structure and performance is central to the study, but not mechanistically supported. The authors should explain why HC was selected over other radical scavengers, and whether the effect is unique to HC. Additional controls with other redox-active molecules are recommended.
The decision to use HC as a molecule to promote interactions with the template and modulate the polymer shell of MMIPs was based on results previously obtained by our group in the development of MMIP systems for environmental applications (unpublished data). In fact, several MMIPs were prepared using dopamine (DA), gallic acid (GA) and HC as antioxidants to be physically or chemically bound to the surface of magnetite. In particular, in the case of DA, its polymerization capacity was exploited to coat the MNPs and obtain functional amino groups that could be used in the interaction with the template. This matrix (Fe3O4@PDA) was compared with systems coated with CS and functionalized with APTES. In the case of PDA, a higher coating was obtained than that of CS (24% vs. 13%, respectively) with an antiradical capacity (AC) of about 13%. This AC value was much lower than those obtained for the Fe3O4@APTES-HCX samples (values reported in this work AC=34-66%). The low AC value allowed the formation of a thicker polymer shell during the MIPs synthesis, as evidenced by TGA measurements (total weight loss of the Fe3O4@PDA-MIP equal to about 40% compared to 14-19% for Fe3O4@APTES-HCX-MIP). Furthermore, to verify the active contribution of an antioxidant molecule other than HC on the radical polymerization of MMA and also to obtain a comparison between two different polymer coatings, gallic acid was chemically bound to CS-coated MNPs. In this case, the Fe3O4@CS-GA0.3 sample showed an AC=59% with a total weight loss in TGA of 22%. It should be noted that approximately 13% of this weight loss was attributable to CS. These results demonstrated the antioxidant's active participation in controlling the thickness of the MIP's polymer shell. Unfortunately, this led to the formation of fewer TBBPA-specific cavities. In fact, this system adsorbed fewer flame retardant molecules (AE=46%). Desorption was also lower, likely due to the greater diffusion of TBBPA into the underlying layers of the CS coating (DE=20%).
For these reasons, HC was chosen, an antioxidant with lower antiradical activity than gallic acid, more suitable for the preparation of systems with adequate adsorption efficiency towards TBBPA, obtained thanks to the possibility of varying the surface content on MNPs.
These results will be the subject of future work, as will the use of other types of antioxidants for the fabrication of new MMIPs with different functionalities.
- The washing and purification steps are critical to remove physically adsorbed (non-grafted) species. However, no convincing evidence is provided that the materials were thoroughly purified. Without this, the observed HC-related effects might be due to unbound HC trapped in the polymer.
We agree with the Reviewer and apologize for the oversight. The washing and UV control procedures for the HC released from the systems, performed at the end of the APTES coupling reaction, have now been reported in the text (Materials and Methods section, subsection 3.5, lines 714-719).
“After removing the nanoparticles with a magnet, they were washed several times with water to eliminate any HC molecules not covalently bound to the MNPs. To verify the effectiveness of this treatment, the washing water was collected and subjected to UV measurements each time until no antiradical activity due to HC released from the systems was observed. The functionalized nanoparticles were then frozen and lyophilized.”
- FTIR spectra do not show convincing evidence of chemical interactions (e.g., no peak shifts or new bands indicating binding). Higher-resolution FTIR or 2D correlation spectroscopy could help resolve this.
Unfortunately, our research group is not equipped to perform measurements of high-resolution FTIR or 2D correlation spectroscopy. However, using standard FTIR spectroscopy, several differences in the spectra of the samples (increase in absorption bands and presence of new bands) could be detected, attributable to the molecules/polymers physically or covalently bound to the MNPs, as reported in subsection 2.1 of the manuscript.
- It is highly recommended to revise the abstract for the good readability. The current structure lacks a clear logical flow, the description is overly dense, and jumps into experimental details too quickly. It should be restructured to emphasize the research problem, proposed solution, key findings, and significance in a coherent order.
As suggested by the Reviewer, the abstract has been revised for the good readability.
- The study does not discuss the colloidal stability or dispersibility of the nanoparticles in aqueous solution. Given that dispersibility affects adsorption efficiency and magnetic recovery, the authors should provide at least one of the following characterizations: the optical image of the dispersion (e.g., to show the Tyndall effect), DLS, zeta potential, or sedimentation data.
As state by the Reviewer, the dispersibility of the nanoparticles in aqueous solution is very important as it affects adsorption efficiency and also magnetic recovery. At this aim, further experiments to verify the stability of the dispersions over time were performed. The results were discussed in the Results and Discussion section (subsection 2.7, lines 514-523) of the manuscript and the obtained images reported in the supplementary information (Figure S2).
Lines 514-523:
“These results could also be attributed to the better dispersibility of the nanoparticles in the Fe3O4@APTES-HC0.7-MIP sample compared to that of the other two systems. Indeed, this sample remained stable for at least 24 hours, showing only a slight aggregation of the nanoparticles at 48 hours (see Figure S2 A and B). With increasing HC concentration, aggregation phenomena were already visible after 24 hours (Fig-ure S2 C). Although the adsorption kinetics were performed while keeping the samples under stirring, the reduced surface/volume ratio due to MNP aggregation could affect the availability of the specific cavities for TBBPA. In fact, although a small variation in adsorption efficiency was observed for the Fe3O4@APTES-HC0.7 systems at 48h (AE=80%), a greater reduction was observed for the Fe3O4@APTES-HC1.0 system (AE=38%).”
- The authors attribute the performance decline over reusability cycles solely to incomplete template desorption. Other factors, such as cavity collapse, matrix aging, or shell degradation, are not considered. The authors should characterize reused particles (e.g., by SEM/FTIR) to verify material stability.
To desorb the template, the Fe3O4@APTES-HC0.7-MIP and Fe3O4@APTES-HC0.7-NIP samples were washed several times with a mixture of acetonitrile:H2O (50:50) before being reused. However, as reported in the manuscript, the template remained entrapped in the organic coating.
To verify this hypothesis and any changes in the morphology of the system, the sample was subjected to SEM-EDX measurements. The SEM micrographs obtained by analyzing the synthesized MMIP sample and the same sample after desorption showed no change in the morphology (no cavity collapse) of the nanoparticles (see Figure S4 A and B). Meanwhile, the EDX measurements confirmed the presence of bromine even after the numerous washings performed on the sample (Figure S4 A and B). This result demonstrated the difficulty of removing TBBPA that had penetrated deeper into the underlying surface layers, probably due to stronger interactions with HC (aromatic steaking interactions).
Therefore, since SEM analysis ruled out possible morphological changes in the samples (cavity collapse) and TGA measurements ruled out possible degradation of the polymer shell, complete removal of the template might be achieved either by using longer washing times or solvents capable of better penetrating the mass of the system.
These newly data and the related discussion are reported in section Results and discussion, subsection 2.10 (lines 639-653):
“To verify the presence of any residual traces of the template that was not removed, SEM-EDX measurements were performed on the sample Fe3O4@APTES-HC0.7-MIP and on the same sample after washing with the acetonitrile:H2O mixture. Figure S4 shows the EDX spectra of these systems (Figure S4 A and B, respectively). As can be seen, the bromine peak was still observable in spectrum B, demonstrating the presence of an unextracted template. By measuring the ratio between the atomic percentage of bromine and the atomic percentage of carbon, it was observed that this value decreased after washings but was still measurable (%Br/%C from 0.40±0.10 to 0.10±0.05). Probably, the traces of the template trapped in the underlying surface layers were due to good interaction established with the antioxidant molecules, which limited its backscattering. Since SEM analysis ruled out possible morphological changes in the samples (Figure S4) and TGA measurements ruled out possible degradation of the polymer shell (same weight loss of the sample before desorption), complete removal of the template might be achieved either by using longer washing times or solvents capable of better penetrating the mass of the system.”

Round 2
Reviewer 1 Report
Comments and Suggestions for Authors
Most of the concerns raised by this referee have been addressed. However, the TGA analysis still requires clarification:
- Please specify how the degradation temperatures were determined. For example, are these values based on the temperature at 5% weight loss?
- As shown in Figure 5, the naked magnetite reaches a thermal plateau between 400–500 °C, indicating its stability within that range. In contrast, the coated samples continue to show significant weight loss at 500 °C, suggesting that residual organic components are still degrading. Therefore, using the weight loss at 500 °C to estimate the total organic content may not be accurate. A higher temperature might be required to fully account for all organic constituents.
Author Response
Reply to Reviewer 1
- Please specify how the degradation temperatures were determined. For example, are these values based on the temperature at 5% weight loss?
The degradation temperature of each sample was determined after calculating the first derivative of each TG curve. For the reviewer's convenience, the TG curves with the corresponding DTG curves of some samples are shown below (Td is represented by the maximum peak of the DTG curve, corresponding to the maximum rate of change in sample mass as a function of temperature).
- As shown in Figure 5, the naked magnetite reaches a thermal plateau between 400–500 °C, indicating its stability within that range. In contrast, the coated samples continue to show significant weight loss at 500 °C, suggesting that residual organic components are still degrading. Therefore, using the weight loss at 500 °C to estimate the total organic content may not be accurate. A higher temperature might be required to fully account for all organic constituents.
We agree with the reviewer that the samples could have been subjected to a wider temperature program, aimed at observing a plateau, in order to be certain of the degradation of the organic material.
However, we felt that it was not very important to determine the absolute degradation rate of the samples, but rather to compare them within a certain temperature range, so that we could deduce which system performed better.
Normally, polymers, even highly crystalline ones, degrade at temperatures below 500 °C. Chitosan is even less stable (Td around 295 °C). As can be seen in the figure above, most of the organic component appears to have degraded before 500 °C. Even if some organic material remained, the further weight loss of the samples would be negligible compared to that recorded previously.

Reviewer 2 Report
Comments and Suggestions for Authors
The authors responses have well addressed my concern, but the figures quality is not good enough to meet the requirement. Please modify the figures carefully according to the policy of the journal .
For example: The figure legends in the manuscript are inconsistent in their use of capitalization. For example, some figures (e.g., Figure 3 and Figure 7) use uppercase letters (A, B, C), while others use lowercase letters (a, b, c). This inconsistency creates confusion and detracts from the overall professionalism of the manuscript. If the figure contains multiple subplots or grouped figures, it is advisable to use other labels such as i), ii), iii)etc.
Comments on the Quality of English Language
There is still minor grammar mistake and inprofessional expression, the authors should check carefully to improve the readability.
Author Response
Reply to Reviewer 2
The authors responses have well addressed my concern, but the figures quality is not good enough to meet the requirement. Please modify the figures carefully according to the policy of the journal.
For example: The figure legends in the manuscript are inconsistent in their use of capitalization. For example, some figures (e.g., Figure 3 and Figure 7) use uppercase letters (A, B, C), while others use lowercase letters (a, b, c). This inconsistency creates confusion and detracts from the overall professionalism of the manuscript. If the figure contains multiple subplots or grouped figures, it is advisable to use other labels such as i), ii), iii etc.
We thank the reviewer for his valuable and helpful suggestions. We apologize for the inaccuracies in the presentation of the figures. These have now been modified in accordance with the journal's policy.
The English has also been revised.

Round 3
Reviewer 1 Report
Comments and Suggestions for Authors
NA